# Mixed-valence molybdenum oxide as a recyclable sorbent for silver removal and recovery from wastewater

Penghui Shao[1], Ziwen Chang[1], Min Li [2]✉, Xiang Lu[1], Wenli Jiang[3], Kai Zhang[1], Xubiao Luo[1] & Liming Yang [1]✉

Silver ions in wastewater streams are a major pollutant and a threat to human health. Given the increasing demand and relative scarcity of silver, these streams could be a lucrative source to extract metallic silver. Wastewater is a complex mixture of many different metal salts, and developing recyclable sorbents with high specificity towards silver ions remains a major challenge. Here we report that molybdenum oxide ($MoO_x$) adsorbent with mixed-valence (Mo(V) and Mo(VI)) demonstrates high selectivity (distribution coefficient of 6437.40 mL g$^{-1}$) for Ag$^+$ and an uptake capacity of 2605.91 mg g$^{-1}$. Our experimental results and density functional theory calculations illustrate the mechanism behind Ag$^+$ adsorption and reduction. Our results show that Mo(V) species reduce Ag$^+$ to metallic Ag, which decreases the energy barrier for subsequent Ag$^+$ reductions, accounting for the high uptake of Ag$^+$ from wastewater. Due to its high selectivity, $MoO_x$ favorably adsorbs Ag$^+$ even in the presence of interfering ions. High selective recovery of Ag$^+$ from wastewater (recovery efficiency = 97.9%) further supports the practical applications of the sorbent. Finally, $MoO_x$ can be recycled following silver recovery while maintaining a recovery efficiency of 97.1% after five cycles. The method is expected to provide a viable strategy to recover silver from wastewater.

Silver (Ag) is among the most important precious metals and has been widely used in various industrial fields, especially the electroplating industry[1]. Massive amounts of Ag$^+$-containing electroplating wastewater are discharged; thus, toxic Ag species are inevitably released into the aquatic environment, causing a large potential risk to the ecological environment and human health[2,3]. In addition, increasing demand for Ag in various fields has caused a crisis, as Ag resources are being depleted[4,5]. Thus, the ability to extract precious Ag from wastewater is of great significance. Numerous technologies have been devoted to recovering Ag$^+$, such as electrochemical deposition[6], membrane separation[7], biological treatment[8], and adsorption[9]. As an economically feasible method, adsorption has attracted significant attention in the remediation of Ag-polluted wastewater[10,11]. Much effort has been dedicated to developing adsorbents with high capacity and excellent selectivity[12–15]. However, due to the interference of coexisting metal ions, the strong acidity of actual Ag$^+$-polluted water, and the widespread dissolved humic acid in the water environment, recovering Ag$^+$ is extremely difficult[16–20]. Furthermore, the complexity of actual Ag$^+$-polluted water has barely been considered in the design and development of Ag$^+$-adsorbents. As a result, the high-performance adsorbents developed in the laboratory are less effective in practical remediation processes.

To selectively recover Ag$^+$, considerable effort has been made to establish the specific interaction between Ag$^+$ and adsorbents, such as

[1]National-Local Joint Engineering Research Center of Heavy Metals Pollutants Control and Resource Utilization, Nanchang Hangkong University, 330063 Nanchang, P. R. China. [2]Department of Chemical Engineering, Chongqing University of Science and Technology, 401331 Chongqing, P. R. China. [3]Key Laboratory of Environmental Biotechnology, Research Center for Eco-Environmental Sciences, Chinese Academy of Sciences, 100085 Beijing, P. R. China. ✉e-mail: limin1406@163.com; yangliming0809185@126.com

                                                                                                    1

ion-imprinting adsorbents and sulfur-rich adsorbents[21,22]. The former method involves selectively adsorbing $Ag^+$ by constructing specific cavities that match $Ag^+$, while the latter method utilizes the very strong ability of sulfur to bind $Ag^+$, which can be attributed to the Lewis soft-soft interactions. Although these materials exhibit excellent selectivity for $Ag^+$ adsorption, the following shortcomings remain: (1) To achieve the recovery of $Ag^+$, these adsorbents need to elute $Ag^+$ through desorbents (such as acid, alkali, or organic solution) or incinerate the adsorbents after adsorption. However, the eluents used cause complicated post-processing procedures and may lead to secondary pollution. Incineration is problematic as the process consumes much energy and generates waste gas. (2) The service life of the adsorbent is also a major obstacle that restricts the application of these materials in actual $Ag^+$-containing wastewater. These adsorbents turn into waste products after use, which not only increases the costs of $Ag^+$ recovery but also contradicts the concept of sustainable development. Therefore, designing a recyclable waste-free adsorbent with the ability to selectively recover $Ag^+$ is a great challenge.

Compared to most metal ions, $Ag^+$ possesses a relatively high redox potential (0.80 V vs. SHE, Supplementary Fig. 1)[23,24] which endows $Ag^+$ with deposition ability on a special reductive adsorbent. Amorphous molybdenum oxide ($MoO_x$), as an excellent photo/electro-catalyst, possesses great electron mass transfer capability[25–28]. The amorphous structure of $MoO_x$ allows it to expose more active sites, which is beneficial to the adsorption of heavy metal ions. Moreover, due to the presence of low-valent Mo in the material, $MoO_x$ exhibits a mild reduction performance. Therefore, it is feasible to recover $Ag^+$ through a spontaneous redox reaction using $MoO_x$ with a mixed valence. In addition, it was found that molybdenum oxide in an ammonia solution could be transformed into ammonium molybdate, which is the raw material for the synthesis of $MoO_x$. This inspired us to design a subversive adsorbent recycling strategy.

Herein, we successfully designed and synthesized an amorphous $MoO_x$ with reductive Mo(V) based on a redox precipitation mechanism using an electrochemical technique. Then, batch $Ag^+$-recovery experiments were performed to evaluate the performance of amorphous $MoO_x$. Through experimental analysis and density functional theory (DFT) calculations, we also fundamentally elucidated the mechanisms of $MoO_x$ capture $Ag^+$. Moreover, a flow-through reactor was designed to evaluate the Ag recovery and demonstrate the superior application potential of $MoO_x$ to recover metallic Ag from actual $Ag^+$-containing wastewater. In addition, a closed-loop recycling method to recover $Ag^+$ and regenerate $MoO_x$ was tested. Finally, we evaluated the regeneration performance of $MoO_x$ and considered the economic benefits of $MoO_x$ recovery Ag to further demonstrate the potential for practical application.

## Results

### Characterization of amorphous $MoO_x$
Amorphous $MoO_x$ was synthesized and loaded onto the F-doped tin oxide (FTO) surface by a simple one-step cyclic voltammetry (CV) electrodeposition method. In contrast to pale green bare FTO (Supplementary Fig. 4a), electrodeposited FTO is covered with a brown $MoO_x$ film. Sheet-like $MoO_x$ with a smooth surface grows uniformly on the FTO surface (Supplementary Fig. 4b). Energy dispersive spectroscopy (EDS) mapping analysis shows that O and Mo are evenly distributed on the surface (Supplementary Fig. 5b, c). The specific surface area of amorphous $MoO_x$ is as low as $17.6\ m^2\ g^{-1}$ (Supplementary Fig. 6a), and the pore size is mainly less than 5 nm, showing a low porosity structure. In the XRD pattern (Supplementary Fig. 6b), all peaks appeared with small peak intensities and wide peak widths, exhibiting an amorphous structure[29], which was also confirmed by the selected area electron diffraction (SAED) pattern (inset of Supplementary Fig. 6b). Compared with regularly arranged crystalline structures, amorphous $MoO_x$ possesses more unsaturated or defective

atoms[30], which are more conducive to capturing heavy metal ions. Furthermore, to ascertain the chemical state of Mo, the corresponding high-resolution Mo $3d$ XPS spectra were analyzed (Supplementary Fig. 6c). Both deconvoluted peaks centered at 231.70 eV and 234.82 eV are assigned to Mo(V), and the other two peaks at 232.70 eV and 235.88 eV correspond to Mo(VI)[31,32]. Impressively, the relative content of Mo(V) is up to 71.9%, while relatively stable Mo(VI) occupies 28.1%. A large amount of Mo(V) atoms is considered to enable the amorphous $MoO_x$ equipped with an excellent redox ability[33,34].

### $Ag^+$ uptake capacity and selectivity
The adsorption isotherm (Fig. 1a) of the amorphous $MoO_x$ demonstrates that the $Ag^+$ uptake capacity is increased promptly with increasing equilibrium concentrations (initial concentrations ranging from 10 to 250 mg $L^{-1}$). For a comparison, bare FTO achieves scarcely any $Ag^+$ removal within 120 min (Supplementary Fig. 7). Two typical isothermal adsorption models (Langmuir and Freundlich models) were used to fit the above data (Supplementary Table 2), and the adsorption process fits better with the Langmuir model ($R^2 = 0.948$). Then, the Langmuir model was employed to predict the theoretical capacity of $MoO_x$ to remove $Ag^+$, and the model has been widely applied to the isotherm data associated with the reduction/oxidation removal/recovery of $As^{3+}$, $Cr^{3+}$, $Au^{3+}$, $Hg^{2+}$, and $Ag^+$[35–39]. Interestingly, the maximum adsorption capacity ($q_m$) was calculated to be as high as 2605.91 mg $g^{-1}$, implying that amorphous $MoO_x$ possesses an excellent application potential in the field of $Ag^+$ recovery.

High adsorbent selectivity is essential for recovering heavy metal ions from actual wastewater[40]. Binary and multi-metal ions mixed solutions were used to evaluate the selectivity of amorphous $MoO_x$. In the binary mixed solution, the $Ag^+$ uptake efficiency is 49.66–473.33 times that of other metal ions at equal initial concentrations (inset of Fig. 1b). In the multi-metal ions mixed solution, as expected, 98.98% of $Ag^+$ is removed; nevertheless, the removal efficiencies of other metal ions are all below 0.3% (Fig. 1b). Besides, the distribution coefficients ($k_d$) of the amorphous $MoO_x$ were also calculated (Supplementary Table 3), and the $k_d$ value for $Ag^+$ is 6437.40 mL $g^{-1}$, which is $4 \times 10^4$ to $2 \times 10^5$ times greater than that of other metal ions (only 0.03–0.15 mL $g^{-1}$). Then, the selectivity coefficients ($k$) of $Ag^+$ relative to the other coexisting ions were also obtained, and the values are all higher than $4.2 \times 10^4$, suggesting that the amorphous $MoO_x$ possesses an excellent selectivity towards $Ag^+$ adsorption.

The $q_m$ and $k$ of the amorphous $MoO_x$ for $Ag^+$ uptake were compared with those of previously reported $Ag^+$-adsorbents (including polymers, ion-imprinted adsorbents, resins, and organic adsorbents, Supplementary Table 4). Figure 1c shows the relationship between $q_m$ and $k$. The $q_m$ value of the amorphous $MoO_x$ is 2.99–73.45 times higher than that of previously reported adsorbents (35.48–872.63 mg $g^{-1}$). In addition to the ultrahigh adsorption capacity, impressively, the amorphous $MoO_x$ has the largest $k$ value (mean value 119866.1), which is approximately 131.82 times that of other adsorbents (only 909.30 of average value). It is well known that ion-imprinted materials possess excellent selectivity; however, the $k$ value of amorphous $MoO_x$ is 1113.58 times that of the ion-imprinted adsorbent (only 107.64). The ultrahigh adsorption capacity and remarkable selectivity endow the amorphous $MoO_x$ with a strong ability to recover Ag from actual Ag-polluted wastewater.

### Anti-interference performance
For amorphous $MoO_x$, the uptake efficiencies of $Ag^+$ are all over 99% under weakly acidic and neutral conditions (Fig. 1d). In addition, the pH value decreases after $Ag^+$ adsorption (e.g., from 6.0 to 3.8), suggesting a proton release process[41]. Notably, even under strongly acidic conditions (Fig. 1d), amorphous $MoO_x$ still maintains an exceptional $Ag^+$ removal efficiency (94.6%), demonstrating superior acid-resistance

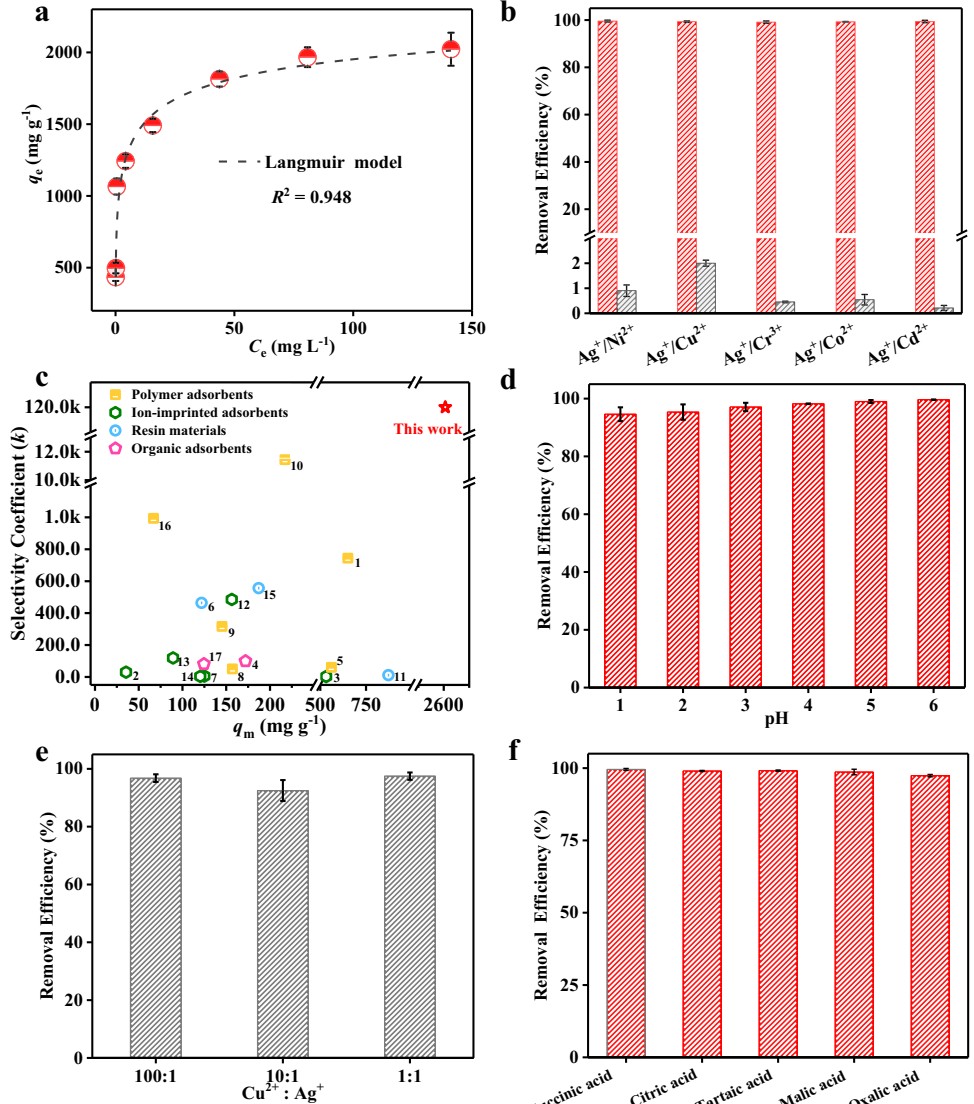

**Fig. 1 | Ag⁺ adsorption and anti-interference performance of MoOₓ. a** Adsorption isotherm of amorphous MoOₓ towards Ag⁺ (initial Ag⁺ concentration was in the range of 10–250 mg L⁻¹, solution volume was 100 mL, and pH was 5.0). The dashed line shows the fitting data of Langmiur model. The functional form and fitting parameters refer to Supplementary Table 2. **b** Removal efficiency for Ag⁺, Ni²⁺, Cu²⁺, Cr³⁺, Co²⁺, and Cd²⁺ in multi-metal and binary-metal (inset) mixed solutions (initial concentration of all metal ions were 20 mg L⁻¹, solution volume was 100 mL, and pH was 5.0). **c** Comparison of the Ag⁺ maximum adsorption capacity ($q_m$) and

selectivity coefficient ($k$) of amorphous MoOₓ with other Ag⁺-adsorbents. **d** Removal efficiency of amorphous MoOₓ for Ag⁺ at different pH. **e** Removal efficiency of Ag⁺ in a binary mixed solution of Ag⁺ and Cu²⁺ (the Cu²⁺/Ag⁺ mass ratio is 1:1, 10:1, and 100:1). **f** Removal efficiency of amorphous MoOₓ for Ag⁺ in the presence of different organic acids (initial Ag⁺ concentration was 20 mg L⁻¹, solution volume was 100 mL, and initial organic acid concentration was 200 mg L⁻¹). All the error bars in this figure represent the standard deviation of the data after two measurements. Source data are provided as a Source Data file.

performance. Furthermore, the selectivity for Ag⁺ was also investigated under high concentrations of other coexisting metal ions. Cu²⁺ was selected for comparison due to the concomitance feature with Ag⁺. High concentrations of Cu²⁺ (the Cu²⁺/Ag⁺ mass ratio up to 100:1) show almost no influence on Ag⁺ adsorption (Fig. 1e), which offers a great possibility of extracting Ag from Cu-containing industrial wastewater. In addition, the effect of salinity was tested using different concentrations of NaNO₃ (from 0 to 1.0 M). More than 97.2% of Ag⁺ is removed under each concentration of NaNO₃, even up to 1.0 M (Supplementary Fig. 8), showing that salt ions in water exhibit little influence on Ag⁺ recovery. Finally, to investigate the interference of dissolved organic matter in wastewater, different small-molecule organic acids with concentrations up to 200 mg L⁻¹ were also studied. Surprisingly, there was no significant change in the removal efficiency of Ag⁺, which remained over 97.5% (Fig. 1f). Amorphous MoOₓ with a nonporous structure and low specific surface area

(17.58 m² g⁻¹) prevents the adsorption of small organic impurities and thus provides a strong resistance to organic pollutant interference[42]. Therefore, amorphous MoOₓ possesses strong adaptability performance, providing feasibility for the selective recovery of Ag⁺ from complex wastewater.

## Self-enhancing recovery mechanism

After the Ag⁺ adsorption process, plenty of tiny silver-white particles accumulate on the surface of the amorphous MoOₓ (Fig. 2a). It is suspected that Ag⁺ is reduced and deposited on MoOₓ. As expected, four typical peaks at $2\theta = 38.2°$, 44.5°, 64.5°, and 77.5°, which are consistent with the crystal phase of metallic Ag appear in the XRD pattern (Fig. 2c)[43]. Wide scan XPS spectra (Supplementary Fig. 9) also suggest the presence of metallic Ag. Furthermore, the high-resolution XPS spectra of Ag 3$d$ were analyzed in detail. Figure 2d shows that the deconvoluted peaks centered at 374.59 eV and 368.50 eV are assigned

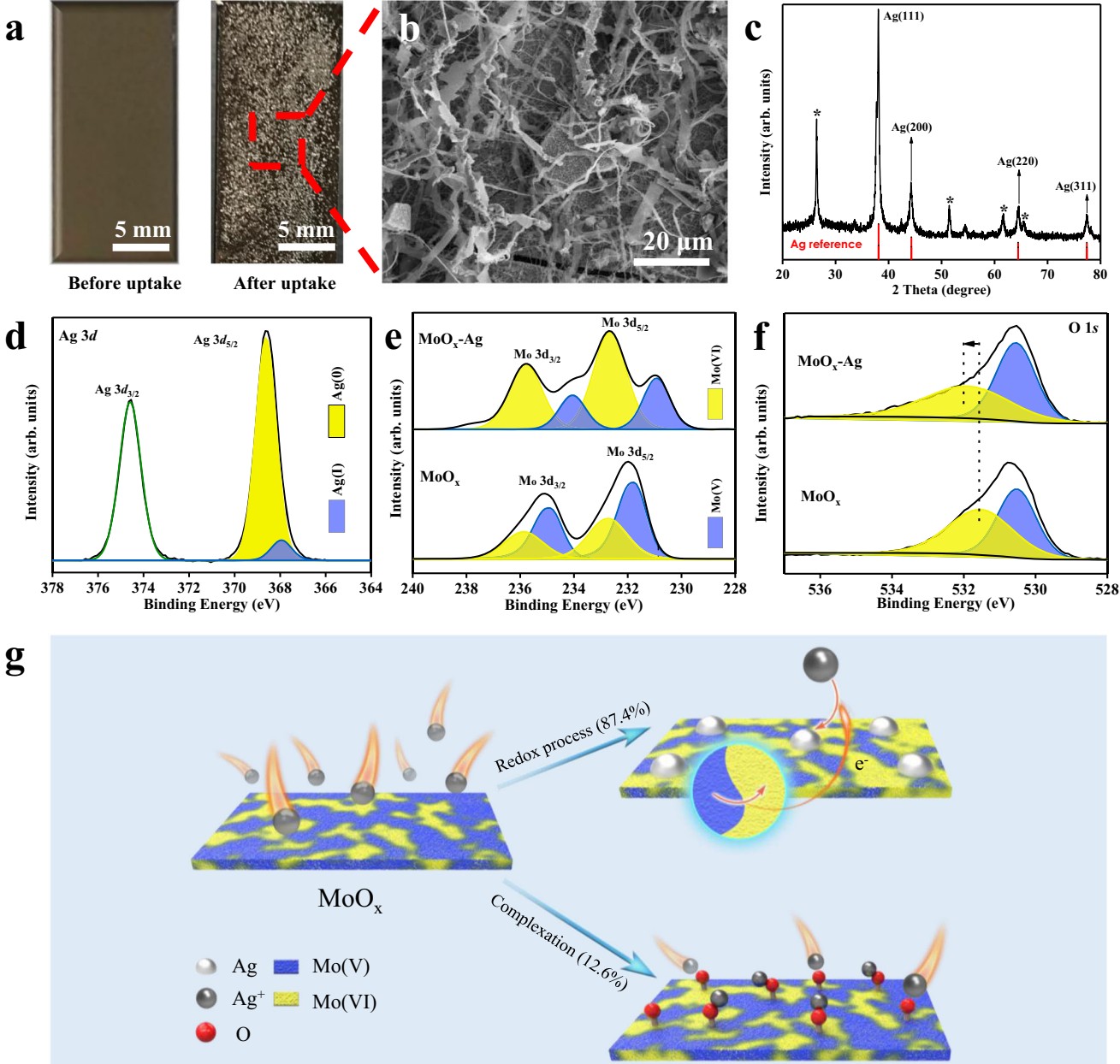

**Fig. 2 | Macroscopic mechanism of Ag⁺ reduction on MoOₓ. a** Optical photographs of FTO loaded with amorphous MoOₓ before and after Ag⁺ adsorption. **b** SEM image revealing strip structure Ag particles. **c** XRD pattern showing the characteristic peaks of metallic Ag (the diffraction peaks marked with "*" originate from the FTO substrate). **d** High-resolution XPS spectra of the Ag 3*d* orbital. High-resolution XPS spectra of **e** Mo 3*d* and **f** O 1*s* orbitals for MoOₓ before and after Ag⁺ uptake. **g** Macroscopic schematics of Ag⁺ capture. Mo(V) on MoOₓ acts as an electron donor to reduce most of Ag⁺, and a small amount of Ag⁺ is trapped by complexation with MoOₓ. Source data are provided as a Source Data file.

to Ag $3d_{5/2}$ and Ag $3d_{3/2}$, respectively[44]. The Ag $3d_{5/2}$ peak was further deconvolved into doublets, which are assigned to Ag(0) and Ag(I)[45,46]. Among the total silver captured by MoOₓ, the proportion of Ag(0) is as high as 87.4%, suggesting that reductive deposition is the dominant way for MoOₓ to capture Ag⁺. Interestingly, the reduced Ag particles form a strip structure and are clustered together (Fig. 2b). This occurs because incipient Ag nanoparticles possess better conductivity (Nyquist plots shown in Supplementary Fig. 10), which could act as an "e-bridge" for transferring electrons from MoOₓ to reduce more outer Ag⁺, and thus, the strip structure was formed[47]. As shown in the EDS mapping images (Supplementary Fig. 11), Ag and Mo are uniformly distributed, suggesting that the reduction deposition of Ag is highly related to Mo on MoOₓ. Moreover, no significant difference was observed for Ag⁺ under light and dark conditions (Supplementary

Fig. 12), demonstrating that the reduction of Ag⁺ is not caused by photo-excited electrons of MoOₓ[48].

To further illuminate the redox process between MoOₓ and Ag⁺, the high-resolution Mo 3*d* XPS spectra were also analyzed after Ag⁺ adsorption (Fig. 2e). The deconvoluted peaks centered at 230.90 eV and 234.08 eV correspond to Mo(V) $3d_{5/2}$ and $3d_{3/2}$, respectively (the other two peaks at 232.70 and 235.88 eV are assigned to Mo(VI))[49]. Interestingly, the relative content of Mo(V) decreased from 71.9 to 28.4% during Ag⁺ deposition; simultaneously, the Mo(VI) content increased from 28.1 to 66.7%. The changes in Mo(V) and Mo(VI) content can be attributed to the oxidation conversion from Mo(V) to Mo(VI), in which the Mo(V) species transfer electrons to realize the reduction of Ag⁺. Notably, after Ag⁺ uptake, the peak position of Mo(V) shifted to lower binding energies by almost 0.9 eV (i.e., from 234.96 to

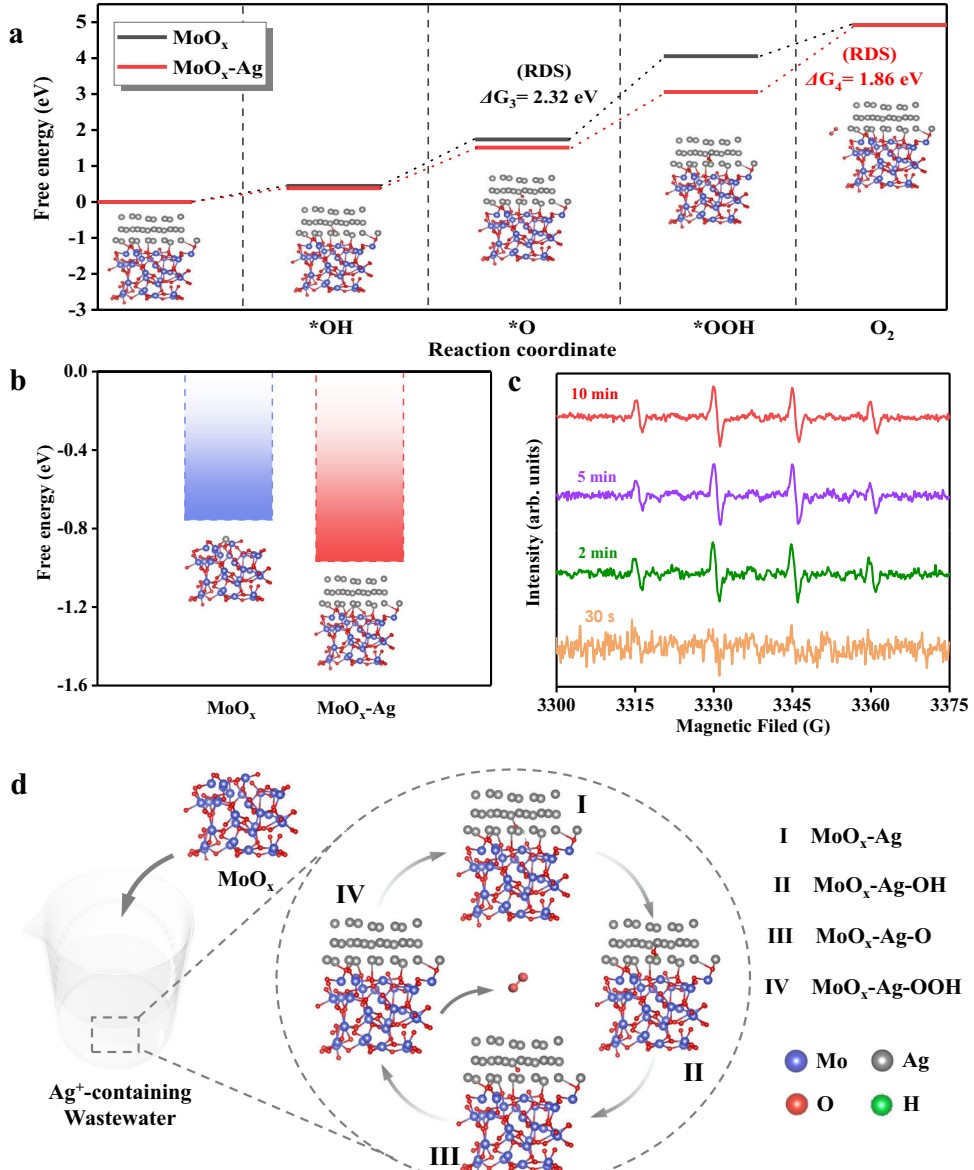

**Fig. 3 | Understanding the self-enhancing reductive deposition mechanism for Ag⁺.** Free energy of **a** the intermediary process of the reduction deposition for Ag⁺ on MoO$_x$ and MoO$_x$-Ag (rate-determining step, RDS), and **b** Ag⁺ reduction deposition on MoO$_x$ and MoO$_x$-Ag. The insets show the optimized MoO$_x$ and intermediates on MoO$_x$-Ag. The data were obtained with DFT calculations. **c** EPR spectra of Ag⁺-capture system after different capture times. **d** Schematics of the self-enhancing reduction mechanism of Ag⁺ on MoO$_x$. The reductive deposition of Ag⁺ on MoO$_x$ is accompanied by a series of intermediate processes (I–IV). All material structure drawings were created by VESTA[57]. Source data are provided as a Source Data file.

234.08 eV for Mo(V) $3d_{3/2}$). This is owing to the interfacial electron transfer between Ag⁺ and Mo(V) species, which could induce the partial structural evolution of external Mo-O[50,51], resulting in a larger binding energy shift. Additionally, high-resolution O 1$s$ XPS spectra before and after Ag⁺ adsorption were also analyzed (Fig. 2f). During Ag⁺ adsorption, the peak position of Mo-O shifted from 531.53 to 531.90 eV, which could result from the complexation interaction between Ag⁺ and O[52]. Based on the above analysis, Ag⁺ uptake by amorphous MoO$_x$ can be mainly attributed to the redox process in which most Ag⁺ (87.4%) was reduced to metallic Ag by the Mo(V) species, at the same time, Mo(V) was converted to Mo(VI) on MoO$_x$. The other part of Ag⁺ (12.6%) was captured by MoO$_x$ through complexation to form Mo-O-Ag and ion exchange reaction with H⁺[53].

Although the reductive deposition of Ag⁺ by MoO$_x$ has been proven to be the main way for Ag⁺ capture on a macroscopic scale, the amount of Mo(V) is lower than that deposited metallic Ag on MoO$_x$.

Considering the inherent excellent electron transport and catalytic ability of silver, it was suspected that the metallic Ag deposited on MoO$_x$ may lower the energy barrier for the reduction deposition of Ag⁺ and strengthen the subsequent reduction process. In this respect, we performed density functional theory (DFT) calculations, simulating the process of Ag⁺ capture by MoO$_x$ and MoO$_x$-Ag (MoO$_x$ deposited metallic Ag). The DFT calculations show that multiple intermediate processes may occur during the reductive deposition of Ag⁺ on MoO$_x$ (Fig. 3a), which is consistent with previous reports[54]. The free energies of *OH, *O, and *OOH (* indicates the active sites on MoO$_x$ and MoO$_x$-Ag) on MoO$_x$-Ag are greater than those on MoO$_x$, which suggests that the deposited Ag improves the surface activity of MoO$_x$. Moreover, the rate-determining step on MoO$_x$ is the transformation of *O → *OOH with a free energy of 2.317 eV, but the rate-determining step on MoO$_x$-Ag changed from *O → *OOH to *OOH → O₂. This is attributed to the stability of *OOH enhanced by the deposition of metallic Ag, which in

turn improved the reactivity of the entire intermediate process[55]. Importantly, $MoO_x$-Ag exhibits a lower free energy of $Ag^+ \rightarrow *Ag$ than $MoO_x$ (−0.756 eV and −0.967 eV, respectively), indicating that $Ag^+$ was more easily reduced and deposited on $MoO_x$-Ag than $MoO_x$ (Fig. 3b). The above results explain why the amount of deposited Ag is higher than the amount of Mo(V) oxidized on $MoO_x$. During the capture of $Ag^+$, the intermediate reactions on the surface of $MoO_x$ can provide additional electrons for subsequent $Ag^+$ reduction, and the deposited Ag on $MoO_x$ lowers the energy barrier for further reductive deposition of $Ag^+$, resulting in more $Ag^+$ being reduced. In-situ EPR was employed to discern the hydroxyl radicals in the $Ag^+$ capture system. Figure 3c shows that hydroxyl radicals produced obvious signals when the adsorption time was 2 min, 5 min, and 10 min. This observation proves that the reductive deposition of $Ag^+$ on $MoO_x$ is accompanied by the above-mentioned intermediate processes and verifies the reliability of the DFT calculation results. Besides, DFT calculation results show that the intermediate process on the $MoO_x$ surface generates protons. This is consistent with the pH change observed in the solution after the adsorption experiment, which also indirectly certifies the accuracy of the DFT calculations.

Integrated with the above experimental and theoretical evidence, it can be concluded that the reduction deposition of $Ag^+$ on $MoO_x$ is a self-enhancing process. Figure 3d shows the self-reinforcing reduction mechanism. In Ag-containing wastewater, Mo(V) on $MoO_x$ acts as an electron donor to donate electrons to reduce $Ag^+$. After metallic Ag is deposited on $MoO_x$, the activity of $MoO_x$ is enhanced, the whole reduction system is more stable, and the energy barrier of subsequent silver ion deposition is lowered, which results in more silver ions being captured. Furthermore, this unique mechanism can well explain the excellent $Ag^+$ capture performance of $MoO_x$. The redox potential of Mo(VI)/Mo(V) is approximately 0.53 V vs. SHE[56] (measured value 0.37 V, Supplementary Fig. 13), much lower than the standard electrode potential of Ag(I)/Ag(0) (about 0.80 V), which enables the redox process spontaneously occur. Nevertheless, competing ions (i.e., $Cu^{2+}$, $Co^{2+}$, $Ni^{2+}$, $Cd^{2+}$, and $Cr^{3+}$) possess lower redox potentials (Supplementary Fig. 1); hence, amorphous $MoO_x$ showed excellent selectivity towards $Ag^+$ uptake. Notably, according to the Nernst equation, $Cu^{2+}$ ions can be removed only if the $Cu^{2+}$ concentration is about $1.13 \times 10^8$ times that of $Ag^+$ (Supplementary Method 4). Consequently, this self-enhancing reduction mechanism could endow amorphous $MoO_x$ with excellent selectivity and strong anti-interference ability for $Ag^+$ recovery from real samples of complex actual wastewater.

### Closed-loop recovery of metallic Ag

Flow-through recovery tests were conducted to investigate the performance of $Ag^+$ capture from real $Ag^+$-containing wastewater samples via the amorphous $MoO_x$. The one-step CV electrodeposition method enables amorphous $MoO_x$ to be loaded on various substrates with different sizes (Supplementary Fig. 14). Low-cost carbon cloth with multiple channels was used for supporting the amorphous $MoO_x$ to act as a membrane implanted in the designed flow-through device (Supplementary Fig. 15). The actual $Ag^+$-containing electroplating wastewater (including COD, $Cu^{2+}$, $Ni^{2+}$, $Zn^{2+}$, $Co^{2+}$, $NO_3^-$, and so on) was pumped through the flow-through device, and the concentrations of effluent were measured. As shown in Fig. 4a, the concentration of $Ag^+$ after filtration was 0.42 mg $L^{-1}$, which can reach the national integrated wastewater discharge standard (GB8978-1996), and the recovery rate reached 0.35 mg $L^{-1}$ $min^{-1}$. Other competing metal ions show no obvious concentration change, which is consistent with the above batch test results. The XRD pattern further verifies the formation of metallic Ag (Fig. 4b), and the purity of Ag recovered from the complex wastewater was as high as 99.79% (details in Supplementary Method 5).

A dilute $NH_3 \cdot H_2O$ solution was employed to recover metallic Ag and realize the regeneration of amorphous $MoO_x$. After dissolution, the metallic Ag particles remained and were collected at the bottom,

enabling Ag recovery. The UV-vis spectra of the regeneration solution are identical to those of the original $(NH_4)_2Mo_2O_7$ (Supplementary Fig. 16), which indicates that the regeneration solution can be used to synthesize amorphous $MoO_x$. The concentrations of recovered $MoO_4^{2-}$ all reach 20 mg $L^{-1}$ in dilute $NH_3 \cdot H_2O$ solution (0.1–0.4 M), and the recovery rate increased with $NH_3 \cdot H_2O$ concentration (Fig. 4c). As expected, amorphous $MoO_x$ was prepared through the above CV electrodeposition technique from the obtained $MoO_4^{2-}$ solution. The regenerated adsorbent can still maintain over 97.1% $Ag^+$ uptake performance even after the fifth cycle (Fig. 4d). Figure 4e shows the schematic of the above closed-loop process of metallic Ag recovery and adsorbent regeneration. Furthermore, an economic analysis of $Ag^+$ recovery was also conducted, and repairing 1 t of this actual $Ag^+$-containing electroplating wastewater resulted in a profit of $552.98 (Supplementary Fig. 17). All these results demonstrate that the amorphous $MoO_x$ can not only recover metallic Ag from complex $Ag^+$-containing wastewater with superior selectivity and ultrahigh anti-interference ability but also realize the sustainable circulation of adsorbents.

## Discussion

In summary, we proposed a strategy for the closed-loop recovery of $Ag^+$ based on amorphous mixed-valence $MoO_x$. The primary mechanism for $Ag^+$ capture on $MoO_x$ was demonstrated to be a self-reinforcing reductive deposition. Mo(V) on $MoO_x$ acts as an electron donor to trigger the reduction of $Ag^+$ and the intermediate processes of oxygen precipitation, which can provide additional electrons for the reduction of $Ag^+$. After the deposition of metallic Ag, the energy barrier for $Ag^+$ reduction is lowered, causing more $Ag^+$ to be reduced on $MoO_x$. Owing to this distinctive mechanism, $MoO_x$ exhibits an ultrahigh capture capacity (2605.91 mg $g^{-1}$), which is among the highest values reported to date, as well as excellent selectivity for $Ag^+$. The recovery of metallic Ag from actual Ag-containing electroplating wastewater was achieved with a purity of up to 99.79%, showing the excellent practical application potential of $MoO_x$. Interestingly, the used amorphous $MoO_x$ can be dissolved in the ammonia solution, and the generated $MoO_4^{2-}$ can be recycled as the raw material for the re-synthesis of $MoO_x$. Compared to conventional solvent desorption, the closed-loop regeneration strategy of $MoO_x$ is waste-free and the capture performance for $Ag^+$ exhibits almost no loss. The $Ag^+$ closed-loop recovery method proposed in this work is potentially meaningful for recycling adsorbents and developing sustainable adsorption technology.

## Methods
### Reagents and materials

Ammonium molybdate tetrahydrate $((NH_4)_6Mo_7O_{24} \cdot 4H_2O, \geq 99\%)$ was purchased from J&K Scientific Ltd. (Shanghai, China). Silver nitrate $(AgNO_3, \geq 99\%)$ was obtained from Aladdin Co., Ltd. (Shanghai, China). Other reagents and materials are described in Supplementary Method 1.

### Preparation of amorphous $MoO_x$

Amorphous $MoO_x$ was synthesized by one-step electrodeposition using the cyclic voltammetry (CV) method. First, 100 mL freshly mixed solution containing 2 mM $(NH_4)_6Mo_7O_{24} \cdot 4H_2O$ (0.2472 g, 0.2 mmol) and 0.5 M $Na_2SO_4$ (7.1 g, 0.05 mol) was prepared to act as the precursor. Then, the CV electrodeposition was carried out under magnetic stirring on a CHI 760E electrochemical workstation (CH Instruments, Shanghai) with a three-electrode system (Supplementary Fig. 2a): A conductive glass FTO as the working electrode, a Pt net as the counter electrode, and a saturated Ag/AgCl electrode as the reference electrode. The potential range was performed between −1.29 and −0.09 V at a scan rate of 50 mV $s^{-1}$. The film was visible after five scans, and after 25 scans, the heights of the two redox peaks approached saturation (Supplementary Fig. 2b). The characterization methods of amorphous $MoO_x$ are presented in Supplementary Method 2.

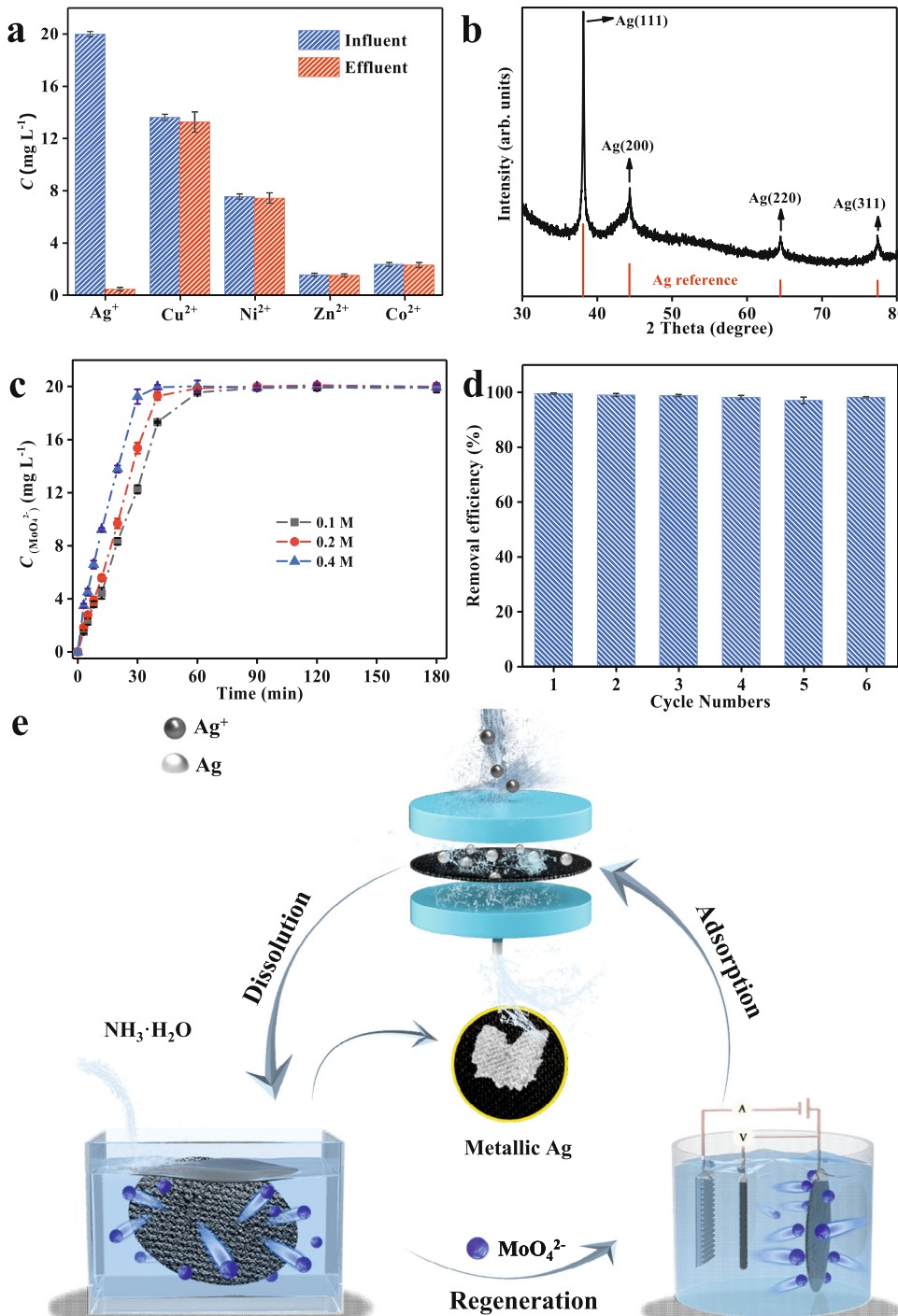

**Fig. 4 | Closed-loop recovery for Ag. a** Concentration changes of different metal ions in actual wastewater samples before and after the flow-through device (solution volume was 100 mL and pH was 5.56). **b** XRD pattern of carbon cloth-loaded amorphous $MoO_x$ after $Ag^+$ uptake. **c** Recovery of $MoO_4^{2-}$ using different concentrations of $NH_3 \cdot H_2O$ (0.1–0.4 M). **d** Removal efficiency of the regenerated amorphous $MoO_x$ for $Ag^+$ (initial $Ag^+$ concentration was 20 mg L$^{-1}$, solution volume was 100 mL, and pH was 5). **e** Schematics of closed-loop recovery of metallic Ag. $MoO_x$ undergoes a cycle of adsorption, dissolution, and regeneration to achieve a closed-loop recovery of Ag. The devices represent the flow-through device (top), the dissolution cell (left), and the electrochemical reactor (right), respectively. All the error bars in this figure represent the standard deviation of the data after two measurements. Source data are provided as a Source Data file.

## Selective adsorption

The adsorption isotherm for $Ag^+$ was investigated with increasing concentrations in batch experiments (details in Supplementary Method 3). For selective adsorption, the coexisting metal ions (i.e., $Ni^{2+}$, $Cu^{2+}$, $Cr^{3+}$, $Cd^{2+}$, and $Co^{2+}$ ions) of the Ag-polluted wastewater were used. A binary solution (composed of $Ag^+$ and the other coexisting ion) and a multi-metal solution consisting of the above six metal ions were used to evaluate the selectivity, respectively. The concentration of each metal ion in the binary and multi-metal solution was 20 mg L$^{-1}$. A piece of amorphous $MoO_x$ (10 mg) was soaked into 100 mL of the above two mixed solutions for 10 h under evenly stirred conditions at room temperature (25 ± 2 °C). Then, the solution was filtered through a syringe filter, and the filtrates were analyzed by an atomic absorption spectrometer (AAS, ContrAA 700, Analytik Jena, Germany) to

determine the concentrations. Finally, the distribution coefficient ($k_d$, mL g$^{-1}$) values were calculated using the following Eq. (1):

$$k_d = \frac{(C_0 - C_e)\,V}{mC_e} \qquad (1)$$

where $C_0$ (mg L$^{-1}$) and $C_e$ (mg L$^{-1}$) are initial and final concentrations, respectively; $V$ (mL) and $m$ (g) is solution volume and adsorbent mass, respectively.

The adsorption selectivity for Ag$^+$ in the presence of other metal ions can be expressed by a selectivity coefficient ($k$) as following Eq. (2):

$$k = \frac{k_{d_1}}{k_{d_2}} \qquad (2)$$

where $k_{d1}$ and $k_{d2}$ are the distribution coefficient (mL g$^{-1}$) of Ag$^+$ and another competing metal ion, respectively.

### Anti-interference tests
The effects of different environmental factors (i.e., pH, salinity, and organic pollutants) on the adsorption performance of the amorphous MoO$_x$ were studied. First, 20 mg L$^{-1}$ AgNO$_3$ aqueous solution was prepared, and HNO$_3$ or NaOH solution was used to adjust the solution pH to the range of 1.0–6.0 (higher pH value would lead to Ag$^+$ precipitation). Amorphous MoO$_x$ was soaked into the above solution for 10 h, and the filtered solution after adsorption was analyzed by AAS. Using the same procedure, salt tolerance of Ag$^+$ adsorption was tested with the addition of NaNO$_3$ solution at different concentrations of 0.001, 0.01, 0.1, and 1.0 M. Moreover, different small-molecule organic acids (succinic acid, citric acid, tartaric acid, malic acid, and oxalic acid) were also tested, and the concentration of the organic acids was 200 mg L$^{-1}$.

### Theoretical calculations
All the calculations were performed within the framework of the density functional theory (DFT) as implemented in the Vienna Ab initio Software Package (VASP 5.4.4) code within the Perdew−Burke−Ernzerhof (PBE) generalized gradient approximation and the projected augmented wave (PAW) method. The cutoff energy for the plane-wave basis set was set to 450 eV. The Brillouin zone of the surface unit cell was sampled by Monkhorst−Pack (MP) grids, with a k-point mesh density of $2\pi \times 0.04$ Å$^{-1}$ for structures optimizations. The convergence criterion for the electronic self-consistent iteration and force was set to 10$^{-5}$ eV and 0.01 eV/Å, respectively. The PBE+U approach was applied to calculations of the electronic structure of MoO$_x$ and MoO$_x$-Ag in this work which can partly reduce the underestimation of the electronic band gap and the excessive tendency to delocalize the electron density. In this work, we set the Hubbard parameter to U − J = 4 eV for Mo. A vacuum layer of 15 Å was introduced to avoid interactions between periodic images.

### Flow-through recovery tests
A flow-through device was designed to realize the Ag$^+$ recovery from actual Ag$^+$-containing electroplating wastewater (Supplementary Fig. 3). The device possesses a chamber with an inner diameter of 3.0 cm and a depth of 2.0 cm. The amorphous MoO$_x$ was loaded onto the surface of carbon cloth (HCP331N, 0.25 mm in thickness, 3.0 cm in radius) through the above CV method using the three-electrode system. Three pieces of amorphous MoO$_x$ modified carbon cloths were stacked in the device chamber and served as a functional filter to trap Ag$^+$. The actual Ag$^+$-containing wastewater was obtained from Nanchang Electroplating Industrial Zone (located at N28°37'84.12″, E116°24'08.45″), and the wastewater characteristics are shown in Supplementary Table 1.

### Closed-loop recovery of metallic Ag
The amorphous MoO$_x$ after Ag$^+$ adsorption was dissolved using different concentrations of NH$_3$·H$_2$O solution (0.1–0.4 M). The concentration of MoO$_4^{2-}$ was measured using a Thermo Scientific ICAP-Q inductively coupled plasma mass spectrometer (ICP-MS, Waltham, MA) and an ultraviolet-visible spectrophotometer (U-3900H, Hitachi, Japan). For the recyclability test, Ag-deposited amorphous MoO$_x$ was immersed in 0.2 M NH$_3$·H$_2$O solution for 2 h. Then, metallic Ag was filtered and recovered, and the MoO$_4^{2-}$ in the filtrate was collected and used as raw materials for the electrochemical synthesis of amorphous MoO$_x$. The regenerated adsorbents were reused in the next cycle of the above flow-through recovery system.

## Data availability
The data that supports the findings of this study are available in the article and Supplementary information file and available from the authors upon request. Source data are provided with this paper.

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

## Acknowledgements

This work was financially supported by the National Natural Science Foundation of China, grant No. 52125002 (X.B.L.), the National Key Research and Development Program of China, grant No. 2019YFC1907900 (X.B.L.), the Natural Science Foundation of Jiangxi Province, grant No. 20224ACB203015 and 20212ACB213006 (P.H.S.), and the Key Project of Research and Development Plan of Jiangxi Province, grant No. 20223BBG74006 and 20201BBE51007 (L.M.Y.).

## Author contributions

L.M.Y., P.H.S., and X.B.L. conceived and supervised the project. P.H.S. and Z.W.C. designed the experiments. X. L. performed experiments on the synthesis of MoOx. Z.W.C. performed the rest of the experiments in this report. P.H.S., Z.W.C., and L.M.Y. analyzed the data and wrote the manuscript. W.L.J. and K.Z. provided constructive suggestions for results and discussion. M.L. and X.B.L. provided constructive suggestions for the manuscript revision. All authors participated in the discussion.

## Competing interests

The authors declare no competing interests.
