## [Peer Review File · Nature Communications]

Mixed-valence molybdenum oxide as a recyclable sorbent for silver removal and recovery from wastewaterREVIEWER COMMENTS

Reviewer #1 (Remarks to the Author):

The authors investigated Ag recovery from wastewater using molybdenum oxide adsorbent to safeguard public health. The authors designed the work systematic way by performing some valuable experimental works accordingly. It is also necessary to critically evaluate new data and not make hasty conclusions that may lead to misinterpretations. However, several points are important to be addressed before going to possible publication in this high-quality journal. Also, the authors need to address all points in the revision stage for a broad range of readers' understanding.

-The English language needs to check carefully in the revision stage because of many careless mistakes in many positions.

-The Figure's quality needs to be improved in the revision stage.

-Abstract: The abstract section is completely different from the Introduction and Experimental sections. The main findings with important opinions are acceptable. The mathematical terms need to be added. The authors need to consider these points in the revision stage.

-References: Many references are not adjacent to this study. The authors need to take notes in the revision stage and cite relevant references including high-impact journals to make the manuscript to a broad range of readers.

-Introduction: There are many studies reported in the literature regarding diverse metal removal and recovery based on different functionalized materials. Composite materials are growing attention for diverse pollutants removal based on their specific functionality. Based on this, do the authors think that the present molybdenum oxide adsorbent is an improvement when compared to other composite materials? The authors need to indicate such points for a broad range of readers. Moreover, the authors need to cite high-impact articles to make the manuscript high-level. The following specific articles may take be noted in the revision stage of Chemical Engineering Journal, 266 (2015) 368–375; Microchemical Journal, 154 (2020) 104585; Chemical Engineering Journal, 307 (2017) 85–94; Chemical Engineering Journal, 324 (2017) 130–139; Journal of Environmental Chemical Engineering, 7 (2019) 103087; Chemical Engineering Journal, 320 (2017) 427–435; Journal of Molecular Liquids, 284 (2019) 502–510; Chemical Engineering Journal, 288 (2016) 368–376; Journal of Environmental Chemical Engineering, 7 (2019) 103378; Composites Part B: Engineering, 171 (2019) 294–301; Journal of Molecular Liquids, 298 (2020) 112035; Chemical Engineering Journal, 259 (2015) 611–619.

-The optimum condition in the removal and recovery operation needs to be determined. The authors need to pay attention in the revision stage.

- In the results and discussion part, the authors only presented the experimental results simply. More detailed analyses are needed to explain why the present molybdenum oxide adsorbent is excellent and how it works. The result and discussion part must be supported in the main manuscript by the following references: Journal of Environmental Chemical Engineering, 7 (2019) 103002; Journal of Molecular Liquids, 296 (2019) 112075; Chemosphere 262 (2021) 127801; Journal of Cleaner Production, 244 (2020) 118805; Chemosphere, 270 (2021) 128668; Journal of Molecular Liquids, 338 (2021) 116667.

- The ion selectivity study needs to be judged as the wastewater containing diverse metal ions.

-The elution/regeneration study needs to be evaluated for the potentiality of the molybdenum oxide adsorbent as a cost-effective.

-Conclusion also needs to be rewritten. Include the following: new concepts and innovations demonstrated in this study, a summary of findings, a comparison with findings by other workers, and a concluding remark.

I would like to see the revised manuscript.

Reviewer #2 (Remarks to the Author):

This paper reports development of a new MoO_x adsorbent for selective Ag(I) recovery from waste water. Systematic studies are performed to reveal its adsorption mechanism and to give practical operation protocols, and sufficient experimental data and numerical simulation results successfully explain the amazing Ag(I) recovery performance. This paper will give worthwhile information to industries dealing with waste water treatment. Originality of the study is well explained by referring relevant articles and previous studies, and comparison in adsorption performance of the new adsorbent with those developed in the previous studies also gives validity of this study. Interpretation of results and discussion seems to be convincing although the reviewer have several questions and comments as following. The manuscript is recommended to be published after revision based on the comments.

1. Adsorption behavior of other monovalent cations should be mentioned.

2. Adsorption of Ag(I) onto MoO₃ is desired to be referred to validate your proposing mechanism. Ag(I) is possible to be adsorbed by ion exchange reaction with H⁺.
3. Influence of anions on the Ag(I) adsorption behavior should be referred.
4. Please give comments on validity of the Langmuir model to the proposed adsorption mechanism (Reduction deposition on MoOx and MoOx-Ag) because adsorption sites are not occupied according to your model.
5. Impurities of the recovered Ag has to be investigated, and discussion on the purity should be added. 97.25 % seems to be insufficient for reuse. Further purification is required or the 2. 75 % impurities will not influence on the applications.
6. Reproducibility of the MoOx production can be shown not only by the Ag(I) recovery ratio but also by the ratio of Mo(V) to Mo(VI)?

Reviewer #3 (Remarks to the Author):

With attention have I read the work "Selective recovery of Ag from wastewater via a waste-free molybdenum oxide adsorbent based on closed REDOX cycle". The paper is written fluently and reports on the application of molybdenum oxide as an effective adsorbent towards silver. The research hypothesis addresses the recovery of silver that is of high importance to sustainable raw material management. The manuscript beholds, however, important flaws, mainly related to: (1) the restricted explanation of the electro-sorption mechanism, considering the capacity exceeds stoichiometric quantity of MoOx, (2) the economical assessment that is conducted in a poor manner, hence compromising the sustainability aspect and (3) the lack of appropriate discussion that frames insights to existing concepts of silver electro-sorption.

Specific comments:

1. The explanation of the reaction mechanism is uncertain, considering the adsorption capacity seems to exceed the reductive capacity of the sorbent. In other words, if the electro-sorption is based on MoOx redox cycling, then only a moderate sorption, below the current maximum is to be expected.
2. L65. The state-of-the-art is incomplete and does not include the possibility for incinerating metal-loaded sorbent.
3. L82-L92: study outcomes should not be listed in the introduction section.
4. Figure 1: too small, recommended to increase the quality of figures.
5. L247: "rate-determining step" and "binding energy" should not be abbreviated.
6. L248: What is the reference for Gibbs free energy with oxygen transformation?

7. L281-285: It is stated that the redox reaction overcomes the potential complexation with organic pollutants, yet such conclusions lack experimental evidence and are speculative.

8. L320 & S5: economical assessment is conceived in a simplistic way. It misses important operating expenditures and is very poorly represented, e.g. electricity is given in weight?

9. L333-L348: The discussion lacks to relate finding with existing understanding of silver electrosorption and does not cite any relevant publications. It is recommended to better interconnect the results with known concepts.

Title: *Selective Recovery of Ag from Wastewater via a Waste-free Molybdenum Oxide Adsorbent Based on Closed REDOX Cycle*

Manuscript ID: NCOMMS-22-43987

We would like to thank the Reviewers for your constructive comments. We have carefully revised our paper according to your comments and **the modifications were marked in red in the revised manuscript**. Enclosed please find the point by point responses to reviewers' comments.

Response to Reviewers' Comments

Reviewer #1 (Remarks to the Author):

General comments: *The authors investigated Ag recovery from wastewater using molybdenum oxide adsorbent to safeguard public health. The authors designed the work systematic way by performing some valuable experimental works accordingly. It is also necessary to critically evaluate new data and not make hasty conclusions that may lead to misinterpretations. However, several points are important to be addressed before going to possible publication in this high-quality journal. Also, the authors need to address all points in the revision stage for a broad range of readers' understanding.*

Response: Many thanks for the Reviewer's valuable comments/suggestions and we accept them all. We have revised the manuscript accordingly, and we hope the revision could be acceptable.

Comment 1. *The English language needs to check carefully in the revision stage because of many careless mistakes in many positions.*

Response: Many thanks for the Reviewer's kind reminder. We carefully checked the entire manuscript and corrected grammatical mistakes under the guidance of professionals. The modifications were marked in red in the revised manuscript.

Comment 2. *The Figure's quality needs to be improved in the revision stage.*

Response: Many thanks for your kind concern. The resolution of the figures in this paper were improved in the revised manuscript.

Comment 3. *Abstract: The abstract section is completely different from the Introduction and Experimental sections. The main findings with important opinions are acceptable. The mathematical terms need to be added. The authors need to consider these points in the revision stage.*

Response: Many thanks for this great suggestion. The abstract was revised in detail. The excellent properties of amorphous MoO_x have been highlighted and more mathematical terms have been added in the abstract. Moreover, the self-reinforcing reduction mechanism of MoO_x for Ag⁺ recovery and the novel regeneration strategy of MoO_x were briefly elucidated. More details were added in the revised manuscripts.

“Achieving the sustainable circulation of adsorbents in Ag⁺ recovery from wastewater with adsorption technology is desirable but challenging. Herein, a waste-free molybdenum oxide (MoO_x) adsorbent with mixed-valence (Mo(V) and Mo(VI)) was designed for selective Ag⁺ recovery from complex wastewater. MoO_x exhibits a remarkable uptake capacity (as high as 2605.91 mg g⁻¹) and an ultrahigh selectivity (distribution coefficient up to 6437.40 mL g⁻¹) for Ag⁺. Even in the presence of different interfering media, the removal efficiency (RE) of MoO_x for Ag⁺ can be maintained above 94%. Density functional theory (DFT) calculations and experimental characterizations unveiled that this exceptional Ag⁺ uptake is due to a self-enhancing reduction mechanism. Specifically, as an electron donor on MoO_x, Mo(V) trends to reduce Ag⁺ to metallic Ag, which decreases the energy barrier for subsequent Ag⁺ reductions. Impressively, MoO_x can realize selective Ag⁺ recovery (RE=97.9%) from actual Ag⁺-containing wastewater. In addition, the raw material (MoO₄²⁻) can be recycled to re-synthesize MoO_x after capturing Ag, realizing closed-loop circulation of adsorbents. Through 5 cycles, the regenerated MoO_x can still maintain over 97.1% Ag⁺ uptake performance. The disruptive adsorbent regeneration strategy with waste-free characteristics reported in this work will be helpful to realize the persistent utilization

of adsorbents.” Please see **Page 2 Lines 21-38**.

Comment 4. *References: Many references are not adjacent to this study. The authors need to take notes in the revision stage and cite relevant references including high-impact journals to make the manuscript to a broad range of readers.*

Response: Thanks very much for this suggestion. We have carried out a meticulous examination of the article references. References that were inconsistent with this paper were deleted and replaced. In the part about mechanism explanation, we cite references from high-impact journals to enhance the convincing power of the article. The relevant modifications were in the revised manuscript and marked in red.

Comment 5. *Introduction: There are many studies reported in the literature regarding diverse metal removal and recovery based on different functionalized materials. Composite materials are growing attention for diverse pollutants removal based on their specific functionality. Based on this, do the authors think that the present molybdenum oxide adsorbent is an improvement when compared to other composite materials? The authors need to indicate such points for a broad range of readers. Moreover, the authors need to cite high-impact articles to make the manuscript high-level. The following specific articles may take be noted in the revision stage of Chemical Engineering Journal, 266 (2015) 368–375; Microchemical Journal, 154 (2020) 104585; Chemical Engineering Journal, 307 (2017) 85–94; Chemical Engineering Journal, 324 (2017) 130–139; Journal of Environmental Chemical Engineering, 7 (2019) 103087; Chemical Engineering Journal, 320 (2017) 427–435; Journal of Molecular Liquids, 284 (2019) 502–510; Chemical Engineering Journal, 288 (2016) 368–376; Journal of Environmental Chemical Engineering, 7 (2019) 103378; Composites Part B: Engineering, 171 (2019) 294–301; Journal of Molecular Liquids, 298 (2020) 112035; Chemical Engineering Journal, 259 (2015) 611–619.*

Response: Many thanks for Reviewer’s constructive comment. As you mentioned, a variety of functional materials have been designed to remove and recover different heavy metal ions in recent years. We consider the amorphous MoO_x adsorbent to be

progressive compared to some current composite adsorbents based primarily on the following points: (1) MoO_x is synthesized in one step by electrochemical method, which is simple and fast. Also, the preparation procedure employs water and hence is nontoxic. (2) The recovery of silver ions on MoO_x is based on a self-enhancing reduction mechanism. This enables MoO_x has an ultra-high Ag⁺ adsorption capacity. Moreover, the self-enhancing mechanism also endows MoO_x with excellent selective adsorption properties, enabling the targeted recovery of Ag⁺ from complex wastewater. (3) Targeting MoO_x, a closed-loop strategy of synthesis-adsorption-dissolution-regeneration is proposed for the first time in this work, which provides a new concept for the persistent utilization of adsorbents. The related discussions were added to the revised manuscript and marked in red.

According to your comments, the relevant references have also been cited in the introduction to enhance the level of this paper.

“Much effort has been dedicated to developing adsorbents with high capacity and excellent selectivity¹²⁻¹⁵.” Please see Page 3 Lines 52-53

Reference:

12. Awual, M. R. New type mesoporous conjugate material for selective optical copper (II) ions monitoring & removal from polluted waters. *Chem. Eng. J.* **307**, 85-94 (2017).
13. Awual, M. R., Hasan, M. M., Znad, H. Organic–inorganic based nano-conjugate adsorbent for selective palladium(II) detection, separation and recovery. *Chem. Eng. J.* **259**, 611-619 (2015).
14. Walden, C., Zhang, W. Biofilms versus activated sludge: Considerations in metal and metal oxide nanoparticle removal from wastewater. *Environ. Sci. Technol.* **50**, 8417-8431 (2016).
15. Asiabi, H., Yamini, Y., Shamsayei, M., Molaei, K., Shamsipur, M. Functionalized layered double hydroxide with nitrogen and sulfur co-decorated carbondots for highly selective and efficient removal of soft Hg²⁺ and Ag⁺ ions. *J. Hazard. Mater.* **357**, 217-225 (2018).

Comment 6. *The optimum condition in the removal and recovery operation needs to be determined. The authors need to pay attention in the revision stage.*

Response: Many Thanks for the Reviewer’s kind reminder. All operating conditions for silver ion removal and recovery were added to the manuscript and the optimal operating conditions were specified in the manuscript. The detailed discussions were added to the manuscript and marked in red.

“Figure 1. Ag⁺ adsorption and anti-interference performance of MoO_x. a

Adsorption isotherm of amorphous MoO_x towards Ag⁺ (initial Ag⁺ concentration was in the range of 10–250 mg L⁻¹; solution volume was 100 mL; and pH was 5.0). **b** Removal efficiency for Ag⁺, Ni²⁺, Cu²⁺, Cr³⁺, Co²⁺, and Cd²⁺ in multi-metal and binary-metal (inset) mixed solutions (initial concentration of all metal ions were 20 mg L⁻¹; solution volume was 100 mL; and pH was 5.0). **c** Comparison of the Ag⁺ maximum adsorption capacity (*q_m*) and selectivity coefficient (*k*) of amorphous MoO_x with other Ag⁺-adsorbents. **d** Removal efficiency of amorphous MoO_x for Ag⁺ at different pH. **e** Removal efficiency of Ag⁺ in a binary mixed solution of Ag⁺ and Cu²⁺ (the Cu²⁺/Ag⁺ mass ratio is 1: 1, 10: 1 and 100: 1). **f** Removal efficiency of amorphous MoO_x for Ag⁺ in the presence of different organic acids (initial Ag⁺ concentration was 20 mg L⁻¹, solution volume was 100 mL and initial organic acid concentration was 200 mg L⁻¹).

Please see **Pages 8-9 Lines 180-191**

“**Figure 4. Closed-loop recovery for Ag.** **a** Concentration changes of different metal ions in actual wastewater samples before and after the flow-through device (solution volume was 100 mL and pH was 5.56). **b** XRD pattern of carbon cloth-loaded amorphous MoO_x after Ag⁺ uptake. **c** Recovery of MoO₄²⁻ using different concentrations of NH₃ H₂O (0.1-0.4 M). **d** Removal efficiency of the regenerated amorphous MoO_x for Ag⁺ (initial Ag⁺ concentration was 20 mg L⁻¹, solution volume was 100 mL and pH was 5). **e** Schematic of closed-loop recovery of metallic Ag.”

Please see **Pages 16-17 Lines 333-339**

Comment 7. *In the results and discussion part, the authors only presented the experimental results simply. More detailed analyses are needed to explain why the present molybdenum oxide adsorbent is excellent and how it works. The result and discussion part must be supported in the main manuscript by the following references: Journal of Environmental Chemical Engineering, 7 (2019) 103002; Journal of Molecular Liquids, 296 (2019) 112075; Chemosphere 262 (2021) 127801; Journal of Cleaner Production, 244 (2020) 118805; Chemosphere, 270 (2021) 128668; Journal of Molecular Liquids, 338 (2021) 116667.*

Response: Many thanks for the Reviewer’s kind concern. We recognized this

shortcoming and carefully revised the results and discussion sections. A more in-depth analysis and discussion has been added to the existing experimental results. The mechanism of silver ion recovery by MoO_x was systematically elucidated. In addition, we have cited the references you mentioned above to confirm the certainty of our work. Specific modifications were marked in red in the revised manuscript.

“High adsorbent selectivity is essential for recovering heavy metal ions from actual wastewater⁴⁰.” Please See Page 6 Lines 132-133

“In addition, the pH value decreases after Ag⁺ adsorption (e.g., from 6.0 to 3.8), suggesting a proton release process⁴¹.” Please See Page 6 Lines 158-159

Reference:

40. Awual, M. R. A facile composite material for enhanced cadmium(II) ion capturing from wastewater. *J. Environ. Chem. Eng.*, **7**: 103378 (2019).
41. Khandaker, S., Chowdhury, M. F., Awual, M. R., Islam, A., Kuba, T. Efficient cesium encapsulation from contaminated water by cellulosic biomass based activated wood charcoal. *Chemosphere.* **262**, 127801 (2021).

Comment 8. *The ion selectivity study needs to be judged as the wastewater containing diverse metal ions.*

Response: Many thanks for your suggestion. For ion selectivity, we constructed binary and multiple heavy metal ion coexistence systems under laboratory conditions to assess the selection of MoO_x for silver ion adsorption. As shown in Figure 1b, the Ag⁺ uptake efficiency is 49.66-473.33 times that of other metal ions at the equal initial concentration in the binary mixed solution. The distribution coefficients (k_d) of MoO_x for Ag⁺ is 6437.40 mL g⁻¹, which is 4×10^4 to 2×10^5 times more than for other metal ions (only 0.03-0.15 mL g⁻¹), proving that the amorphous MoO_x possesses an excellent selectivity towards Ag⁺ adsorption. Besides, the flow-through recovery tests are conducted to further investigate the Ag⁺ capture performance from actual Ag⁺-containing electroplating wastewater. In Figure 4a, the concentration of Ag⁺ after filtration was 0.42 mg L⁻¹, yet, the other competing metal ions shows no obvious concentration change. This results demonstrate that MoO_x maintains excellent Ag⁺ selectivity even in actual wastewater.

Figure 1. b Removal efficiency for Ag⁺, Ni²⁺, Cu²⁺, Cr³⁺, Co²⁺, and Cd²⁺ in multi-metal and binary-metal (inset) mixed solutions (initial concentration of all metal ions were 20 mg L⁻¹; solution volume was 100 mL; and pH was 5.0).

Figure 4. a Concentration changes of different metal ions in actual wastewater samples before and after the flow-through device (solution volume was 100 mL and pH was 5.56).

Comment 9. *The elution/regeneration study needs to be evaluated for the potentiality of the molybdenum oxide adsorbent as a cost-effective.*

Response: Many thanks to the Reviewer for the constructive suggestion. The cost involved in the regeneration of MoO_x has been added to the original economic accounting. Meanwhile, the economic benefits of MoO_x for the recovery of Ag⁺ have been re-calculated, which demonstrates the potential of MoO_x to be applied as a cost-effective adsorbent. The detailed calculations were presented in the *Supplementary*

Information (Figure S17).

Figure S17. Economic analysis of Ag⁺ recovery from 1 t Ag⁺-containing wastewater.

Comment 10. Conclusion also needs to be rewritten. Include the following: new concepts and innovations demonstrated in this study, a summary of findings, a comparison with findings by other workers, and a concluding remark.

Response: Many thanks for the Reviewer's great advice and the section of discussion has been rewritten. In the revised discussion, we highlighted the new strategy of MoO_x closed-looped Ag⁺ recovery, briefly described the innovation of the self-enhanced reduction mechanism for MoO_x capture Ag⁺, and summarized the excellent adsorption performance as well as the strong anti-interference of MoO_x. Besides, MoO_x is considered to be one of the best adsorbents for Ag⁺ capture by comparison with other works. Finally, the application of MoO_x and the close-looped recovery strategies in the field of precious metal recovery was prospected. More details

were added in the revised Manuscript and marked in red.

“Discussion

In summary, we proposed a strategy for the closed-loop recovery of Ag^+ based on amorphous mixed-valence MoO_x . The primary mechanism for Ag^+ capture on MoO_x was demonstrated to be a new self-reinforcing reductive deposition. The Mo(V) on MoO_x acts as an electron donor to trigger the reduction of Ag^+ and the intermediate processes of oxygen precipitation, which can provide additional electrons for the reduction of Ag^+ . After the deposition of metallic Ag, the energy barrier for Ag^+ reduction is lowered, causing more Ag^+ to be reduced on MoO_x . Owing to this distinctive mechanism, MoO_x exhibits an extremely high capture capacity (2605.91 mg g^{-1}), which is among the highest values reported to date, as well as excellent selectivity for Ag^+ . The recovery of metallic Ag from actual Ag-containing electroplating wastewater was achieved with a purity of up to 99.79%, showing the excellent practical application potential of MoO_x . To our delight, the used amorphous MoO_x can be dissolved in the ammonia solution and the generated MoO_4^{2-} can be recycled as the raw material for the re-synthesis of MoO_x . Compared to conventional solvent desorption, the closed-loop regeneration strategy of MoO_x is waste-free and the capture performance for Ag^+ exhibits almost no loss. The Ag^+ closed-loop recovery method proposed in this work is potentially meaningful for recycling adsorbents and developing sustainable adsorption technology.” **Please see Page 17 Lines 341-358**

I would like to see the revised manuscript.

Reviewer #2 (Remarks to the Author):

General comments: *This paper reports development of a new MoO_x adsorbent for selective Ag(I) recovery from waste water. Systematic studies are performed to reveal its adsorption mechanism and to give practical operation protocols, and sufficient experimental data and numerical simulation results successfully explain the amazing Ag(I) recovery performance. This paper will give worthwhile information to industries dealing with waste water treatment. Originality of the study is well explained by referring relevant articles and previous studies, and comparison in adsorption performance of the new adsorbent with those developed in the previous studies also gives validity of this study. Interpretation of results and discussion seems to be convincing although the reviewer have several questions and comments as following. The manuscript is recommended to be published after revision based on the comments.*

Response: Many thanks to the Reviewer for the very positive comments and valuable suggestions, which are much helpful to improve the scientific merits of the manuscript.

Comment 1. *Adsorption behavior of other monovalent cations should be mentioned.*

Response: Many thanks for the Reviewer's comment. Based on your suggestion, two binary solution (composed of Ag⁺/Na⁺, Ag⁺/K⁺ and Ag⁺/Li⁺) and a multi-metal solution consisted of the above four metal ions was used to evaluate the effect of coexisting monovalent cations on Ag⁺ adsorption, respectively. The concentrations of all the metal ions were 20 mg L⁻¹. A piece of amorphous MoO_x was soaked into 100 mL of the above solutions for 10 h under evenly stirred condition at room temperature (25 ± 2 °C), respectively. Then, the solution was filtered through a syringe filter, and the filtrates were analyzed by an atomic absorption spectrometer (AAS, ContrAA 700, Analytik Jena, Germany) to determine the concentrations of Ag⁺, Na⁺, K⁺ and Li⁺.

As shown in the figure below, MoO_x almost exclusively extracts Ag (above 99.5%) but not the other three monovalent cations (below 0.2%) either in binary or multiple coexistence systems, indicating that the presence of monovalent metal cations does not

interfere to the adsorption of Ag^+ on MoO_x . This is in line with our expectations. Because, Ag^+ have a higher reduction potential and are more easily reduced to metallic Ag on MoO_x , compared with other monovalent metal cations. Attributed to the unique reduction recovery mechanism of MoO_x , it can overcome the interference of monovalent metal cations and selectively reduce and recover Ag^+ from wastewater.

Figure R1. Removal efficiency for Ag^+ , Na^+ , K^+ and Li^+ in **a** binary-metal and **b** multi-metal mixed solution (initial concentration of all metal ions were 20 mg L^{-1} ; solution volume was 100 mL ; and pH was 5.0).

Comment 2. Adsorption of Ag(I) onto MoO_3 is desired to be referred to validate your proposing mechanism. Ag(I) is possible to be adsorbed by ion exchange reaction with H^+ .

Response: Many thanks for your great comment. By researching the literature relating to MoO_3 adsorbents, there is a significant difference in the mechanism of Ag^+ adsorption on MoO_x and MoO_3 . The chemical valence of Mo on MoO_x includes Mo(V) and Mo(VI), which gives MoO_x the ability to reduce Ag^+ from wastewater. As seen in the XRD pattern of MoO_x after Ag^+ adsorption (Figure 2c), four typical peaks which are consistent with the crystal phase of metallic Ag appear. From the high-resolution XPS spectra of Ag 3d (Figure 2d), the proportion of metallic Ag is as high as 87.4% among the total silver captured by MoO_x . Based on high-resolution XPS spectra of MoO_x O1s before and after adsorption (Figure 2f) and the relevant molybdenum oxide adsorbent literature, the other part of Ag^+ (12.6%) is considered to be captured by MoO_x through complexation to form Mo-O-Ag and ion exchange reaction with H^+ . Therefore, the reductive deposition was demonstrated to be the dominant way for MoO_x to capture

Ag⁺. While, the other small percentage of Ag⁺ are captured by complexation and ion exchange. The related modifications were added in the manuscript and marked in red.

“To further illuminate the redox process between MoO_x and Ag⁺, the high-resolution Mo 3d XPS spectra were also analysed after Ag⁺ adsorption (Figure 2e). The deconvoluted peaks centred at 230.90 eV and 234.08 eV correspond to Mo(V) 3d_{5/2} and 3d_{3/2}, respectively (the other two peaks at 232.70 eV and 235.88 eV are assigned to Mo(VI))⁴⁹. Interestingly, the relative content of Mo(V) decreased from 71.9% to 28.4% during Ag⁺ deposition, simultaneously, the Mo(VI) content increased from 28.1% to 66.7%. The changes in Mo(V) and Mo(VI) content can be attributed to the oxidation conversion from Mo(V) to Mo(VI), in which the Mo(V) species transfer electrons to realize the reduction of Ag⁺. Notably, after Ag⁺ uptake, the peak position of Mo(V) shifted to lower binding energies (BE) by almost 0.9 eV (i. e., from 234.96 eV to 234.08 eV for Mo(V) 3d_{3/2}). This is owing to the interfacial electron transfer between Ag⁺ and Mo(V) species, which could induce the partial structural evolution of external Mo-O^{50, 51}, resulting in a larger BE shift. Additionally, high-resolution O 1s XPS spectra before and after Ag⁺ adsorption were also analysed (Figure 2f). During Ag⁺ adsorption, the peak position of Mo-O shifted from 531.53 eV to 531.90 eV, which could result from the complexation interaction between Ag⁺ and O⁵². **Based on the above analysis, Ag⁺ uptake by amorphous MoO_x can be mainly attributed to the redox process in which most Ag⁺ (87.4%) was reduced to metallic Ag by the Mo(V) species, at the same time, Mo(V) was converted to Mo(VI) on MoO_x. The other part of Ag⁺ (12.6%) was captured by MoO_x through complexation to form Mo-O-Ag and ion exchange reaction with H⁺⁵³.” Please see **Pages 9-10 Lines 213-233****

Figure 2. c XRD pattern showing the characteristic peaks of metallic Ag (the diffraction peaks marked with * come from FTO substrate).

Figure 2. d High-resolution XPS spectra of Ag 3d orbital for MoO_x after Ag^+ capture.

Figure 2. f High-resolution XPS spectra of O 1s orbital for MoO_x before and after Ag⁺ uptake.

References:

53. Li, Y., Shaheen, S. M., Azeem, M., Zhang, L., Feng, C., Peng, J., Qi, W., Liu, J., Luo, Y., Peng, Y., Ali, E. F., Smith, K., Rinklebe, J., Zhang, Z., Li, R. Removal of lead (Pb²⁺) from contaminated water using a novel MoO₃-biochar composite: Performance and mechanism. *Environ. Pollut.* **308**, 119693 (2022).

Comment 3. *Influence of anions on the Ag(I) adsorption behavior should be referred.*

Response: Many thanks to you for this great suggestion. In order to investigate the influence of different anions on Ag⁺ uptake, two binary solutions (composed of Ag⁺/Cr⁶⁺ and Ag⁺/Sb⁵⁺) and a multi-metal solution consisting of the above three metal ions was prepared, respectively. The concentrations of all the metal ions were 20 mg L⁻¹. A piece of amorphous MoO_x was soaked into 100 mL of the above solutions for 10 h under evenly stirred condition at room temperature (25 ± 2 °C), respectively. Then, the solution was filtered through a syringe filter, and the filtrates were analyzed by an atomic absorption spectrometer (AAS, ContrAA 700, Analytik Jena, Germany) to determine the concentrations of Ag⁺, Cr⁶⁺ and Sb⁵⁺.

From Figure R2, the presence of heavy metal oxygenated anions almost has no effect on the capture of Ag⁺. MoO_x achieves over 98.5% removal efficiency for Ag⁺, both in binary and multi-metal co-existence systems. Furthermore, the effect of different concentrations of NO₃⁻ on the capture of Ag⁺ was also evaluated. As seen in Figure S8, more than 97.2% of Ag⁺ is removed under each concentration of NaNO₃ even up to 1.0 mol L⁻¹, showing NO₃⁻ in water has little influence on the Ag⁺ recovery. In summary, the presence of anions in the water hardly interferes with the efficient capture for Ag⁺ of MoO_x.

Figure R2. Removal efficiency for Ag^+ , Cr^{6+} and Sb^{5+} in **a** binary-metal and **b** multi-metal mixed solution (initial concentration of all metal ions were 20 mg L^{-1} ; solution volume was 100 mL; and pH was 5.0).

Figure S8. Removal efficiency of amorphous MoO_x for Ag^+ in different concentrations of nitrate.

Comment 4. Please give comments on validity of the Langmuir model to the proposed adsorption mechanism (Reduction deposition on MoO_x and $\text{MoO}_x\text{-Ag}$) because adsorption sites are not occupied according to your model.

Response: Many thanks for the Reviewer's comment. In this work, the Langmuir model was employed to predict the theoretical Ag^+ removal capacity of MoO_x , and it has been widely applied to model the isotherm data associated with the reduction/oxidation removal/recovery of As^{3+} , Cr^{3+} , Au^{3+} , Hg^{2+} , and Ag^+ .

The capture of silver ions by MoO_x is considered as a process of adsorption followed by reductive deposition. In the first stage of Ag^+ extraction, silver ions are uniformly adsorbed onto the MoO_x surface, which is a monolayer adsorption, in accordance with the Langmuir model. Immediately afterwards, the adsorbed Ag^+ are reduced on MoO_x and the original adsorption sites are released to capture more silver

ions. Moreover, the deposited metallic Ag can effectively reduce the energy barrier for subsequent Ag^+ deposition, making subsequent Ag^+ reduction easier, which is the reason for the ultra-high Ag^+ adsorption capacity of MoO_x . The relevant details of the modifications were added to the revised manuscript and marked in red.

“The adsorption isotherm (Figure 1a) of the amorphous MoO_x demonstrates that the Ag^+ uptake capacity is increased promptly with increasing equilibrium concentrations (initial concentrations ranging from 10 mg L^{-1} to 250 mg L^{-1}). For a comparison, bare FTO achieves scarcely any Ag^+ removal within 120 min (Figure S7). Two typical isothermal adsorption models (Langmuir and Freundlich models) were used to fit the above data (Table S2), and the adsorption process fits better with the Langmuir model ($R^2 = 0.948$). Then, the Langmuir model was employed to predict the theoretical capacity of MoO_x to remove Ag^+ , and the model has been widely applied to the isotherm data associated with the reduction/oxidation removal/recovery of As^{3+} , Cr^{3+} , Au^{3+} , Hg^{2+} , and Ag^+ ³⁵⁻³⁹. To our delight, the maximum adsorption capacity (q_m) was calculated to be as high as $2605.91 \text{ mg g}^{-1}$, implying that amorphous MoO_x possesses an outstanding application potential in the field of Ag^+ recovery.” Please see

Page 5 Lines 120-131

References:

35. Yuan, M., Yao, H., Xie, L., Liu, X., Wang, H., Islam, S. M., Shi, K., Yu, Z., Sun, G., Li, H., Ma, S., Kanatzidis, M. G. Polypyrrole- Mo_3S_{13} an Efficient Sorbent for the Capture of Hg^{2+} and Highly Selective Extraction of Ag^+ over Cu^{2+} . *J. Am. Chem. Soc.* **142**, 1574-1583 (2020).
36. Wang, Z. Y., Sim, A., Urban, J. J., Mi, B. X. Removal and recovery of heavy metal ions by two-dimensional MoS_2 nanosheets: performance and mechanisms. *Environ. Sci. Technol.* **52**, 9741-9748 (2018).
37. Zuo, K., Huang, X., Liu, X., Gil Garcia, E. M., Kim, J., Jain, A., Chen, L., Liang, P., Zepeda, A., Verduzco, R., Lou, J., Li, Q. A Hybrid Metal-Organic Framework-Reduced Graphene Oxide Nanomaterial for Selective Removal of Chromate from Water in an Electrochemical Process. *Environ. Sci. Technol.* **54**, 13322-13332 (2020).
38. Hong, Y., Thirion, D., Subramanian, S., Yoo, M., Choi, H., Kim, H. Y., Stoddart, J. F., Yavuz, C. T. Precious metal recovery from electronic waste by a porous porphyrin polymer. *Proc. Natl. Acad. Sci. U.S.A.* **117**, 16174-16180 (2020).
39. Fu, K., Liu, X., Yu, D., Luo, J., Wang, Z., Crittenden, J. C. Highly efficient and selective Hg (II) removal from water using multilayered $\text{Ti}_3\text{C}_2\text{O}_x$ MXene via adsorption coupled with catalytic reduction mechanism. *Environ. Sci. Technol.* **54**, 16212-16220 (2020).

Comment 5. Impurities of the recovered Ag has to be investigated, and discussion on the purity should be added. 97.25 % seems to be insufficient for reuse. Further purification is required or the 2.75 % impurities will not influence on the applications.

Response: Many thanks for Reviewer's great suggestion. The purity of the recovered Ag^+ is calculated based on the change in concentration of all metal ions in the wastewater. We believe that in the material regeneration stage, ammonia will not only dissolve the MoO_x adsorbent, but also dissolve the impurity ions adsorbed on MoO_x (Cu^{2+} , Zn^{2+} , Ni^{2+} , and Co^{2+}), so as to obtain higher purity metallic Ag. On this basis, we detected the concentration of heavy metal ions in the regeneration solution and recalculated the purity of the recovered Ag. 50 mL 0.2 mol L^{-1} of ammonia solution was used to dissolve MoO_x and impurity metal ions. As seen in Figure R3, trace amounts of Ag^+ and most of impurity ions are dissolved into the ammonia solution. This result confirms that the purity of the recovered Ag has been improved. Combined with the data in Figure 4a, the purity of the recovery Ag was recalculated to be as high as 99.79%, which has a high recovery value. The section of the manuscript relating to the purity of the recovered Ag has been revised accordingly and marked in red. The details for calculating the purity of recovered Ag^+ were listed in the *Supplementary Information (Text S5)*.

Figure R3. The concentration of Ag^+ , Cu^{2+} , Ni^{2+} , Zn^{2+} and Co^{2+} in the $\text{NH}_3 \text{ H}_2\text{O}$ solution after the dissolution of MoO_x .

Text S5

The purity of the recovered metallic Ag can be calculated by the following formula:

$$p = \frac{m_1}{m}$$

(Equation S3)

Where the p is the purity of recovered Ag (%), m_1 is the mass of recovered Ag (mg), and m is the total mass of recovered metals (including Ag^+ , Cu^{2+} , Ni^{2+} , Zn^{2+} , and Co^{2+} , mg).

The values of m_1 and m are obtained from the difference in the concentration of heavy metal ions in the wastewater and regenerative solutions, where the volume of the wastewater and regenerative solutions is 100 mL and 50 mL, respectively. Through calculation, the values of m_1 and m are 1.955 and 1.959 mg, respectively. Therefore, the purity of silver recovered from wastewater is as high as 99.79%.

Comment 6. *Reproducibility of the MoO_x production can be shown not only by the Ag(I) recovery ratio but also by the ratio of Mo(V) to Mo(VI)?*

Response: Many thanks for the Reviewer's valuable comment. Following your suggestion, the high-resolution Mo 3d XPS spectra after 6 regenerations were analysed to demonstrate the superiority of closed-looped regeneration of MoO_x (Figure R4). Impressively, the ratio of Mo(V) to Mo(VI) on MoO_x after 6 regenerations is 2.1, which is similar to the ratio on the original MoO_x (2.5). Therefore, MoO_x can still maintain the original chemical structure and excellent Ag⁺ capture performance after 6 regenerations, which proves that it has a good reproducibility.

Figure R4. High-resolution XPS spectra of Mo 3d orbitals for MoO_x and MoO_x after 5 cycles.

Reviewer #3 (Remarks to the Author):

General comments: *With attention have I read the work “Selective recovery of Ag from wastewater via a waste-free molybdenum oxide adsorbent based on closed REDOX cycle”. The paper is written fluently and reports on the application of molybdenum oxide as an effective adsorbent towards silver. The research hypothesis addresses the recovery of silver that is of high importance to sustainable raw material management. The manuscript beholds, however, important flaws, mainly related to: (1) the restricted explanation of the electro-sorption mechanism, considering the capacity exceeds stoichiometric quantity of MoO_x, (2) the economical assessment that is conducted in a poor manner, hence compromising the sustainability aspect and (3) the lack of appropriate discussion that frames insights to existing concepts of silver electro-sorption.*

Response: We appreciate the Reviewer for the very positive comments on the quality and novelty of this manuscript. The great comments are all accepted and carefully addressed as below.

Comment 1. *The explanation of the reaction mechanism is uncertain, considering the adsorption capacity seems to exceed the reductive capacity of the sorbent. In other words, if the electro-sorption is based on MoO_x redox cycling, then only a moderate sorption, below the current maximum is to be expected.*

Response: Many thanks for the Reviewer’s great comment. In this work, electrochemistry is used only as a means of material synthesis and regeneration. The process of capturing Ag⁺ by MoO_x is not electrically charged which is different from electro-sorption.

Based on XRD, SEM and XPS characterization before and after MoO_x capture of Ag⁺, it can be proved that the main mechanism of MoO_x capture of Ag⁺ is reduction deposition. Then, the theoretical calculation was employed to in-depth analyze this mechanism. The DFT calculations show that there may be multiple intermediate processes in the reductive deposition of Ag⁺ on MoO_x. To our delight, the deposited of

metallic Ag can effectively reduce the free energy of these intermediate reactions, allowing these processes to be easier. These intermediate reactions can provide additional energy to reduce silver ions on MoO_x. Furthermore, the deposited metallic Ag on MoO_x effectively reduces the energy required for the reduction of Ag⁺, which caused more silver ions to be reduced. This result well explains why the amount of deposited Ag is higher than the amount of Mo(V) oxidized on MoO_x. In addition, the in-situ EPR spectra indicate the production of hydroxyl radicals during Ag⁺ deposition, verified the reliability of the DFT calculation results. Therefore, it can be concluded that the recovery mechanism of Ag⁺ on MoO_x is mainly a self-enhancing reductive deposition. The detailed discussions were in the revised manuscript (**Pages 9-11**).

Comment 2. L65. *The state-of-the-art is incomplete and does not include the possibility for incinerating metal-loaded sorbent.*

Response: Many thanks for your kind reminder. We have revised the manuscript accordingly and marked it in red.

“The former method involves selectively adsorbing Ag⁺ by constructing specific cavities that match Ag⁺, while the latter method utilizes the very strong ability of sulfur to bind Ag⁺, which can be attributed to the Lewis soft-soft interactions. Although these materials exhibit excellent selectivity for Ag⁺ adsorption, the following shortcomings remain: (i) To achieve the recovery of Ag⁺, these adsorbents need to elute Ag⁺ through desorbents (such as acid, alkali or organic solution) or incinerate the adsorbents after adsorption. However, the eluents used cause complicated post-processing procedures and may lead to secondary pollution. Incineration is problematic as the process consumes much energy and generates waste gas.” **Please see Page 3 Lines 66-70**

Comment 3. L82-L92: *study outcomes should not be listed in the introduction section.*

Response: Many thanks for your kind concern. We have removed the statement about the study outcomes and revised the introduction. The details were added in the revised manuscript and marked in red.

“Herein, we successfully designed and synthesized an amorphous MoO_x with

reductive Mo(V) based on redox precipitation mechanism using an electrochemical technique. Then, batch Ag^+ -recovery experiments were performed to evaluate the performance of amorphous MoO_x . Through experimental analysis and density functional theory (DFT) calculations, we also fundamentally elucidated the mechanisms of MoO_x capture Ag^+ . Moreover, a flow-through reactor was designed to evaluate the Ag recovery and demonstrate the superior application potential of MoO_x to recover metallic Ag from actual Ag^+ -containing wastewater. In addition, a closed-loop recycling method to recover Ag^+ and regenerate MoO_x was tested. Finally, we evaluated the regeneration performance of MoO_x and considered the economic benefits for MoO_x recovery Ag to further demonstrate the potential for practical application.”

Please see Page 4 Lines 87-97

Comment 4. *Figure 1: too small, recommended to increase the quality of figures.*

Response: Thank you very much for your suggestion. We have made adjustments to Figure 1 and the resolution of all images in this paper has been improved.

Figure 1. Ag^+ adsorption and anti-interference performance of MoO_x . **a** Adsorption isotherm of amorphous MoO_x towards Ag^+ (initial Ag^+ concentration was in the range of 10–250 $mg\ L^{-1}$; solution volume was 100 mL; and pH was 5.0). **b** Removal efficiency for Ag^+ , Ni^{2+} , Cu^{2+} , Cr^{3+} , Co^{2+} , and Cd^{2+} in multi-metal and binary-metal (inset) mixed solutions (initial concentration of all metal ions were 20 $mg\ L^{-1}$; solution volume was 100 mL; and pH was 5.0). **c** Comparison of the Ag^+ maximum adsorption capacity (q_m) and selectivity coefficient (k) of amorphous MoO_x with other Ag^+ -adsorbents. **d** Removal efficiency of amorphous MoO_x for Ag^+ at different pH. **e** Removal efficiency of Ag^+ in a binary mixed solution of Ag^+ and Cu^{2+} (the Cu^{2+}/Ag^+ mass ratio is 1: 1, 10: 1 and 100: 1). **f** Removal efficiency of amorphous MoO_x for Ag^+

in the presence of different organic acids (initial Ag^+ concentration was 20 mg L^{-1} , solution volume was 100 mL and initial organic acid concentration was 200 mg L^{-1}).

Comment 5. L247: “rate-determining step” and “binding energy” should not be abbreviated.

Response: Many thanks for the Reviewer’s kind reminder. We modified all these abbreviations to their full names. The modifications were marked in red in the revised manuscript.

“The **free energies** of $*\text{OH}$, $*\text{O}$, and $*\text{OOH}$ ($*$ indicates the active sites on MoO_x and $\text{MoO}_x\text{-Ag}$) on $\text{MoO}_x\text{-Ag}$ are greater than those on MoO_x , which suggests that the deposited Ag improves the surface activity of MoO_x . Moreover, the rate-determining step (RDS) on MoO_x is the transformation of $*\text{O} \rightarrow *\text{OOH}$ with a **free energy** of 2.317 eV , but the **rate-determining step** on $\text{MoO}_x\text{-Ag}$ changed from $*\text{O} \rightarrow *\text{OOH}$ to $*\text{OOH} \rightarrow \text{O}_2$. **This attributed to the stability of $*\text{OOH}$ enhanced by the deposition of metallic Ag , which in turn improved the reactivity of the entire intermediate process⁵⁵.** Importantly, $\text{MoO}_x\text{-Ag}$ exhibits a lower **free energy** of $\text{Ag}^+ \rightarrow *\text{Ag}$ than MoO_x (-0.756 and -0.967 eV , respectively), indicating that Ag^+ was more easily reduced and deposited on $\text{MoO}_x\text{-Ag}$ than MoO_x (**Figure 3b**).” **Please see Page 12 Lines 251-261**

Comment 6. L248: What is the reference for Gibbs free energy with oxygen transformation?

Response: Many thanks for the Reviewer’s valuable comment. This oxygen transformation intermediate reaction has been demonstrated in electrochemical oxygen evolution. By analyzing the results of the DFT calculations, the reductive deposition of Ag^+ on MoO_x was demonstrated to be accompanied by an intermediate process of oxygen evolution in this work. The trend in Gibbs free energy changes with oxygen transformation is similar to that previously reported in the literature.

In our work, the Mo(V) on MoO_x acts as an electron donor, allowing a series of intermediate processes to be carried out. The stability of this reaction process was increased after the deposition of metallic Ag . Moreover, these intermediate processes

can generate additional electrons to provide for the reduction of Ag^+ . On the other hand, the energy barrier for the reduction of Ag^+ on MoO_x was decreased after Ag deposition, resulting in more Ag^+ being reduced on MoO_x . More discussions were added to the revised manuscript and marked in red. In addition, relevant literature has been cited to better prove the DFT calculation results.

“The DFT calculations show that multiple intermediate processes may occur during the reductive deposition of Ag^+ on MoO_x (Figure 3a), which is consistent with previous reports⁵⁴ The free energies of *OH, *O, and *OOH (* indicates the active sites on MoO_x and $\text{MoO}_x\text{-Ag}$) on $\text{MoO}_x\text{-Ag}$ are greater than those on MoO_x , which suggests that the deposited Ag improves the surface activity of MoO_x . Moreover, the rate-determining step (RDS) on MoO_x is the transformation of $*\text{O}\rightarrow*\text{OOH}$ with a free energy of 2.317 eV, but the rate-determining step on $\text{MoO}_x\text{-Ag}$ changed from $*\text{O}\rightarrow*\text{OOH}$ to $*\text{OOH}\rightarrow\text{O}_2$. This attributed to the stability of *OOH enhanced by the deposition of metallic Ag, which in turn improved the reactivity of the entire intermediate process⁵⁵. Importantly, $\text{MoO}_x\text{-Ag}$ exhibits a lower free energy of $\text{Ag}^+\rightarrow*\text{Ag}$ than MoO_x (-0.756 and -0.967 eV, respectively), indicating that Ag^+ was more easily reduced and deposited on $\text{MoO}_x\text{-Ag}$ than MoO_x (Figure 3b). The above results explain why the amount of deposited Ag is higher than the amount of Mo(V) oxidized on MoO_x . Please see Page 12 Lines 249-262

References

54. Li, X., Wang, Y., Wang, J., Da, Y., Zhang, J., Li, L., Zhong, C., Deng, Y., Han, X., Hu, W.. Sequential electrodeposition of bifunctional catalytically active structures in $\text{MoO}_3/\text{Ni-NiO}$ composite electrocatalysts for selective hydrogen and oxygen evolution. *Adv. Mater.* **32**, 2003414 (2020).
55. Zhang, H., Zhou, W., Dong, J., Lu, X. F., Lou, X. W. D. Intramolecular electronic coupling in porous iron cobalt (oxy)phosphide nanoboxes enhances the electrocatalytic activity for oxygen evolution. *Energ. Environ. Sci.* **12**, 3348-3355 (2019).

Comment 7. L281-285: *It is stated that the redox reaction overcomes the potential complexation with organic pollutants, yet such conclusions lack experimental evidence and are speculative.*

Response: Many thanks for your kind reminder. We were aware of the inaccuracy

of this statement. The sentence has been deleted in the revised Manuscript.

Comment 8. L320 & S5: *economical assessment is conceived in a simplistic way. It misses important operating expenditures and is very poorly represented, e.g. electricity is given in weight?*

Response: Many thanks for the Reviewer’s suggestion and we accept it. The economic benefits of MoO_x for the recovery of Ag⁺ have been re-calculated in the revised Manuscript. We have considered the consumption of three processes: synthesis of MoO_x, water treatment, recovery and regeneration. The detailed calculation was presented in the *Supplementary Information (Figure S17)*.

Figure S17. Economic analysis of Ag⁺ recovery from 1 t Ag⁺-containing wastewater.

Comment 9. L333-L348: *The discussion lacks to relate finding with existing understanding of silver electrosorption and does not cite any relevant publications. It*

is recommended to better interconnect the results with known concepts.

Response: Many thanks for the Reviewer's constructive advice. It should be noted that the electrochemistry was used only as a means of material synthesis and regeneration in this work. The process of capturing Ag^+ on MoO_x was not electrically charged which is different from electro-sorption. Based on your suggestions, we combined the findings from our work with the existing understanding of Ag^+ adsorption and cited relevant literature in an attempt to better integrate the results and known concepts. In addition, we will later carry out research on the electrosorption for Ag^+ on MoO_x , to expand the applicability of MoO_x . More details were added in the revised Manuscript and marked in red.

“Discussion

In summary, we proposed a strategy for the closed-loop recovery of Ag^+ based on amorphous mixed-valence MoO_x . The primary mechanism for Ag^+ capture on MoO_x was demonstrated to be a new self-reinforcing reductive deposition. The Mo(V) on MoO_x acts as an electron donor to trigger the reduction of Ag^+ and the intermediate processes of oxygen precipitation, which can provide additional electrons for the reduction of Ag^+ . After the deposition of metallic Ag, the energy barrier for Ag^+ reduction is lowered, causing more Ag^+ to be reduced on MoO_x . Owing to this distinctive mechanism, MoO_x exhibits an extremely high capture capacity (2605.91 mg g^{-1}), which is among the highest values reported to date, as well as excellent selectivity for Ag^+ . The recovery of metallic Ag from actual Ag-containing electroplating wastewater was achieved with a purity of up to 99.79%, showing the excellent practical application potential of MoO_x . To our delight, the used amorphous MoO_x can be dissolved in the ammonia solution and the generated MoO_4^{2-} can be recycled as the raw material for the re-synthesis of MoO_x . Compared to conventional solvent desorption, the closed-loop regeneration strategy of MoO_x is waste-free and the capture performance for Ag^+ exhibits almost no loss. The Ag^+ closed-loop recovery method proposed in this work is potentially meaningful for recycling adsorbents and developing sustainable adsorption technology.” **Please see Page 17 Lines 341-358**

REVIEWERS' COMMENTS

Reviewer #1 (Remarks to the Author):

The authors are not serious about revision. The authors need to seriously revise the manuscript to get acceptance from this high-quality journal. A major revision is required.

The authors investigated Ag recovery from wastewater using molybdenum oxide adsorbent to safeguard public health. The authors designed the work systematic way by performing some valuable experimental works accordingly. It is also necessary to critically evaluate new data and not make hasty conclusions that may lead to misinterpretations. However, several points are important to be addressed before going to possible publication in this high-quality journal. Also, the authors need to address all points in the revision stage for a broad range of readers' understanding.

-The English language needs to check carefully in the revision stage because of many careless mistakes in many positions.

-The Figure's quality needs to be improved in the revision stage.

-Abstract: The abstract section is completely different from the Introduction and Experimental sections. The main findings with important opinions are acceptable. The mathematical terms need to be added. The authors need to consider these points in the revision stage.

-References: Many references are not adjacent to this study. The authors need to take notes in the revision stage and cite relevant references including high-impact journals to make the manuscript to a broad range of readers.

-Introduction: There are many studies reported in the literature regarding diverse metal removal and recovery based on different functionalized materials. Composite materials are growing attention for diverse pollutants removal based on their specific functionality. Based on this, do the authors think that the present molybdenum oxide adsorbent is an improvement when compared to other composite materials? The authors need to indicate such points for a broad range of readers. Moreover, the authors need to cite high-impact articles to make the manuscript high-level. The following specific articles may take be noted in the revision stage of Chemical Engineering Journal, 266 (2015) 368–375; Microchemical Journal, 154 (2020) 104585; Chemical Engineering Journal, 307 (2017) 85–94; Chemical Engineering Journal, 324 (2017) 130–139; Journal of Environmental Chemical Engineering, 7 (2019) 103087; Chemical Engineering Journal, 320 (2017) 427–435; Journal of Molecular Liquids, 284 (2019) 502–510;

Chemical Engineering Journal, 288 (2016) 368–376; Journal of Environmental Chemical Engineering, 7 (2019) 103378; Composites Part B: Engineering, 171 (2019) 294–301; Journal of Molecular Liquids, 298 (2020) 112035; Chemical Engineering Journal, 259 (2015) 611–619.

-The optimum condition in the removal and recovery operation needs to be determined. The authors need to pay attention in the revision stage.

- In the results and discussion part, the authors only presented the experimental results simply. More detailed analyses are needed to explain why the present molybdenum oxide adsorbent is excellent and how it works. The result and discussion part must be supported in the main manuscript by the following references: Journal of Environmental Chemical Engineering, 7 (2019) 103002; Journal of Molecular Liquids, 296 (2019) 112075; Chemosphere 262 (2021) 127801; Journal of Cleaner Production, 244 (2020) 118805; Chemosphere, 270 (2021) 128668; Journal of Molecular Liquids, 338 (2021) 116667.

- The ion selectivity study needs to be judged as the wastewater containing diverse metal ions.

-The elution/regeneration study needs to be evaluated for the potentiality of the molybdenum oxide adsorbent as a cost-effective.

-Conclusion also needs to be rewritten. Include the following: new concepts and innovations demonstrated in this study, a summary of findings, a comparison with findings by other workers, and a concluding remark.

I would like to see the revised manuscript.

Reviewer #2 (Remarks to the Author):

The manuscript was appropriately revised based on my questions and comments to Authors. I do not have any additional comments to the revised manuscript.

Reviewer #3 (Remarks to the Author):

1. General comments:

Efforts of the authors to improve the discussion are appreciated and lead to a better interconnection of the findings with existing scientific concepts. Certain explanations however reflect a limited understanding of fundamental chemical processes and could have been refined at a more detailed level. In particular, this relates the description of reaction enthalpy and activation energy.

While the material and application mechanism can definitely be subject for further studying, it is perceived as sufficiently documented in the proposed manuscript.

2. Unresolved issues:

- L205: "because"

- Figure S1: The oxidation state notation of Mo^{6+} is invalid and should be Mo(VI) as the element evidently does not prevail as a hexavalent cation.

- L222 and L254: The rate determining step and, respectively, binding energy are still abbreviated in the text, while this is uncommon.

Title: *Mixed-valence molybdenum oxide as a recyclable sorbent for silver removal and recovery from wastewater*

Manuscript ID: NCOMMS-22-43987A

We would like to thank the Reviewers for your constructive comments. We have carefully revised our paper according to your comments and **the modifications were marked in red in the revised manuscript**. Enclosed please find the point by point responses to reviewers' comments.

Response to Reviewers' Comments

Reviewer #1 (Remarks to the Author):

General comments: *The authors are not serious about revision. The authors need to seriously revise the manuscript to get acceptance from this high-quality journal. A major revision is required. The authors investigated Ag recovery from wastewater using molybdenum oxide adsorbent to safeguard public health. The authors designed the work systematic way by performing some valuable experimental works accordingly. It is also necessary to critically evaluate new data and not make hasty conclusions that may lead to misinterpretations. However, several points are important to be addressed before going to possible publication in this high-quality journal. Also, the authors need to address all points in the revision stage for a broad range of readers' understanding.*

Response: We appreciate the reviewer repeated comments, but we believe all these points have been raised and addressed in the previous round of review.

Reviewer #2 (Remarks to the Author):

General comments: *The manuscript was appropriately revised based on my questions and comments to Authors. I do not have any additional comments to the revised manuscript.*

Response: Many thanks to the Reviewers for the recognition of this work.

Reviewer #3 (Remarks to the Author):

General comments: *Efforts of the authors to improve the discussion are appreciated and lead to a better interconnection of the findings with existing scientific concepts. Certain explanations however reflect a limited understanding of fundamental chemical processes and could have been refined at a more detailed level. In particular, this relates the description of reaction enthalpy and activation energy.*

While the material and application mechanism can definitely be subject for further studying, it is perceived as sufficiently documented in the proposed manuscript.

Response: We appreciate the Reviewer for the positive comments on this work, and we also thank you for pointing out the shortcomings in our previous round of responses. Based on your suggestions, we reinterpreted the self-reinforcing reduction mechanism of Ag^+ on MoO_x .

According to the XRD, SEM and XPS characterizations of MoO_x after captured Ag^+ , it can be demonstrated that Mo(V) on MoO_x acts as an electron donor for the reductive deposition of Ag^+ . Considering the inherent excellent electron transport and catalytic ability of silver, it was suspected that the metallic Ag deposited on MoO_x may strengthen the reduction of subsequent Ag^+ ions on MoO_x . In this respect, we performed density functional theory (DFT) calculations, simulating the process of Ag^+ capture by MoO_x and $\text{MoO}_x\text{-Ag}$ (MoO_x deposited metallic Ag). The DFT calculations show that multiple intermediate processes may occur during the reductive deposition of Ag^+ on MoO_x ($\text{MoO}_x\text{-OH}$, $\text{MoO}_x\text{-O}$, and $\text{MoO}_x\text{-OOH}$). Moreover, the free energies of these intermediate processes on $\text{MoO}_x\text{-Ag}$ are greater than MoO_x , indicating that these processes are easier to perform on MoO_x after metallic Ag deposited. The electrons generated from these intermediate processes are also able to supply additional Ag^+ to reduction on MoO_x . Furthermore, the deposited metallic Ag on MoO_x effectively reduces the energy barrier for the reduction of Ag^+ , which caused more silver ions to be reduced. The in-situ EPR spectra indicate the production of hydroxyl radicals during Ag^+ deposition, verifying the reliability of the DFT calculation results. Therefore, it can be conclude that the recovery mechanism of Ag^+ on MoO_x is mainly a self-enhancing

reductive deposition.

This work investigates the performance, mechanism, and application of MoO_x to capture silver ions. However, the recovery other precious metal ions (such as gold, platinum and palladium) on MoO_x has yet to be verified and studied. In addition, molybdenum oxide materials possess great photoelectric properties. The combination of photo/electrochemistry with MoO_x for the recovery of precious metals also deserves further study.

Comment 1. L205: "because"

Response: Many thanks for the Reviewer's kind concern. We fixed this spelling error in revised manuscript and marked in red.

"This occurs **because** incipient Ag nanoparticles possess better conductivity (Nyquist plots shown in Figure S10), which could act as an "e-bridge" for transferring electrons from MoO_x to reduce more outer Ag⁺, and thus, the strip structure was formed⁴⁷." Please see **Page 8 Line 193**

Comment 2. Figure S1: The oxidation state notation of Mo⁶⁺ is invalid and should be Mo(VI) as the element evidently does not prevail as a hexavalent cation.

Response: Many thanks for the Reviewer's kind reminder. We recognized this error and the oxidation state notation in Figure S1 has been corrected.

Supplementary Figure 1. Standard redox potential diagrams for different metal ions (vs. SHE).

Comment 3. L222 and L254: *The rate determining step and, respectively, binding energy are still abbreviated in the text, while this is uncommon.*

Response: Many thanks to Reviewer for the kind reminder. We carefully examined the manuscript and modified all these abbreviations to their full names. The modifications were marked in red in the revised manuscript.

“Notably, after Ag^+ uptake, the peak position of Mo(V) shifted to lower binding energies by almost 0.9 eV (i. e., from 234.96 eV to 234.08 eV for Mo(V) $3d_{3/2}$). This is owing to the interfacial electron transfer between Ag^+ and Mo(V) species, which could induce the partial structural evolution of external Mo-O^{50, 51}, resulting in a larger binding energy shift.”

“Moreover, the rate-determining step on MoO_x is the transformation of $*\text{O} \rightarrow *\text{OOH}$ with a free energy of 2.317 eV, but the rate-determining step on $\text{MoO}_x\text{-Ag}$ changed from $*\text{O} \rightarrow *\text{OOH}$ to $*\text{OOH} \rightarrow \text{O}_2$.” Please see **Pages 8-9 Lines 213-234**